# Mechanistic insights into the active site and allosteric communication pathways in human nonmuscle myosin-2C

**Krishna Chinthalapudi[1,2,3†], Sarah M Heissler[1,4†], Matthias Preller[1,5], James R Sellers[4]\*, Dietmar J Manstein[1,2]\***

[1]Institute for Biophysical Chemistry, OE4350, Hannover Medical School, Hannover, Germany; [2]Division for Structural Biochemistry, OE8830, Hannover Medical School, Hannover, Germany; [3]Cell Adhesion Laboratory, Department of Integrative Structural and Computational Biology, The Scripps Research Institute, Jupiter, United States; [4]Laboratory of Molecular Physiology, NHLBI, National Institutes of Health, Bethesda, United States; [5]Centre for Structural Systems Biology (CSSB), German Electron Synchrotron (DESY), Hamburg, Germany

**Abstract** Despite a generic, highly conserved motor domain, ATP turnover kinetics and their activation by F-actin vary greatly between myosin-2 isoforms. Here, we present a 2.25 Å pre-powerstroke state (ADP·VO$_4$) crystal structure of the human nonmuscle myosin-2C motor domain, one of the slowest myosins characterized. In combination with integrated mutagenesis, ensemble-solution kinetics, and molecular dynamics simulation approaches, the structure reveals an allosteric communication pathway that connects the distal end of the motor domain with the active site. Disruption of this pathway by mutation of hub residue R788, which forms the center of a cluster of interactions connecting the converter, the SH1-SH2 helix, the relay helix, and the lever, abolishes nonmuscle myosin-2 specific kinetic signatures. Our results provide insights into structural changes in the myosin motor domain that are triggered upon F-actin binding and contribute critically to the mechanochemical behavior of stress fibers, actin arcs, and cortical actin-based structures.
DOI: https://doi.org/10.7554/eLife.32742.001

**\*For correspondence:**
sellersj@nhlbi.nih.gov (JRS);
manstein.dietmar@mh-hannover.de (DJM)

[†]These authors contributed equally to this work

**Competing interests:** The authors declare that no competing interests exist.

## Introduction

The extent to which filamentous actin (F-actin) can activate the enzymatic output of conventional myosins-2 varies by more than two orders of magnitude (*Heissler and Sellers, 2016*; *Sellers, 2000*). Despite a substantial sequence identity and a highly conserved actomyosin ATPase cycle, structural and allosteric adaptations causative for the tremendous enzymatic and hence physiological differences are largely unknown (*Figure 1A*) (*Heissler and Sellers, 2016*, *2013*).

Nonmuscle myosin-2C, the gene product of *MYH14*, is one of the slowest myosins-2 as its steady-state ATPase activity lacks potent F-actin activation (*Heissler and Manstein, 2011*). Transient kinetic signatures of the nonmuscle myosin-2C enzymatic cycle include a high affinity for ADP, small kinetic ($k_{-AD}/k_{-D}$) and thermodynamic ($K_{AD}/K_D$) coupling ratios, and a higher duty ratio than it is commonly found in myosins-2 (*Figure 1A*, *Figure 1—figure supplement 1A*). Together, these signatures qualify nonmuscle myosin-2C as a dynamic strain-sensing actin tether (*Heissler and Manstein, 2013*; *Heissler and Manstein, 2011*; *Bloemink and Geeves, 2011*). The participation in the active regulation of cytoplasmic contractility in cellular processes including cytokinesis, neuronal dynamics, adhesion, and tension maintenance are in line with this interpretation (*Takaoka et al., 2014*; *Wylie and Chantler, 2008*; *Ebrahim et al., 2013*). Nonmuscle myosin-2C has recently received great attention as the near atomic resolution cryo electron microscopic structure of its motor domain bound to

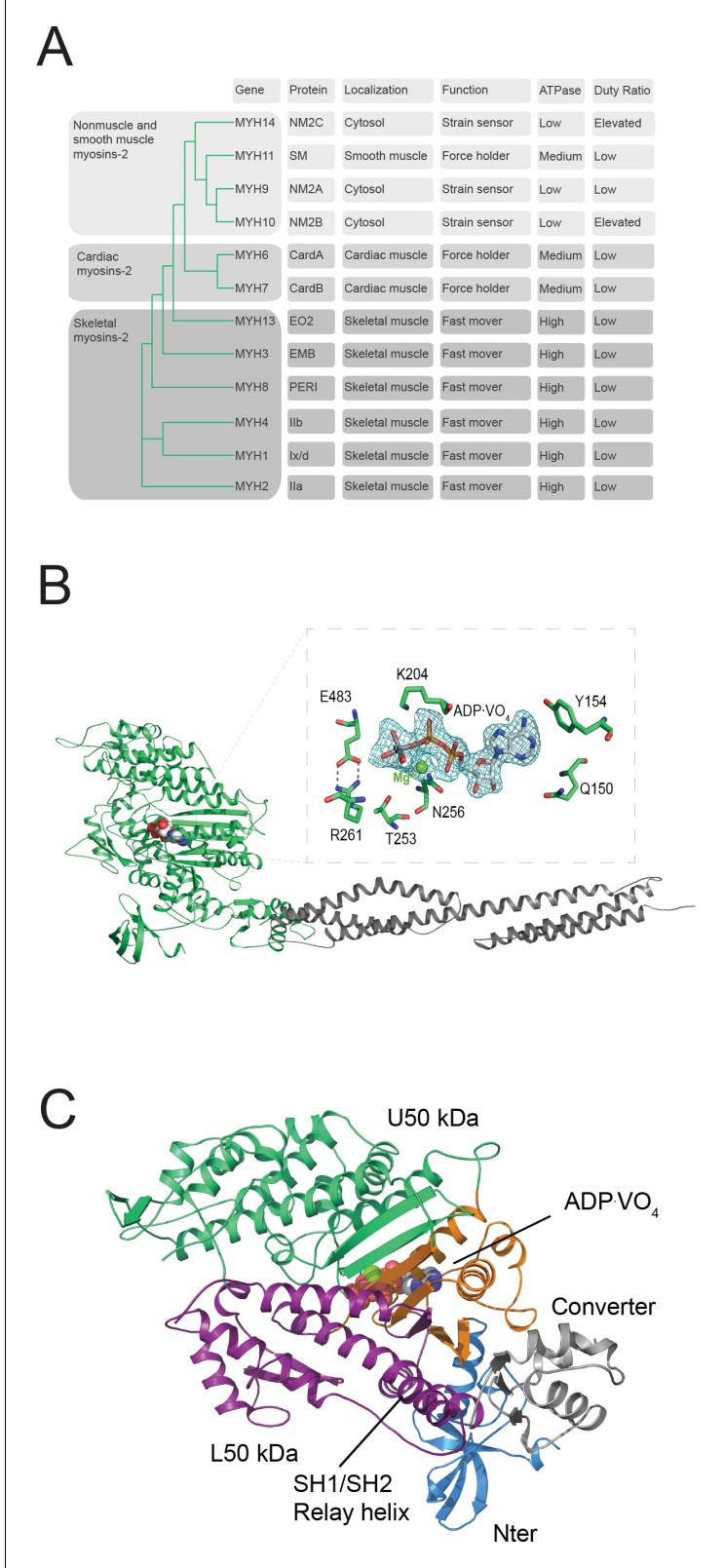

**Figure 1.** Myosin-2 phylogeny, overall topology, and active site characteristics of human NM2C. (**A**) Phylogenetic analysis divides human myosin-2 in the three subfamilies (i) nonmuscle and smooth muscle myosins-2, (ii) cardiac, (iii) and skeletal muscle myosins-2 (*Foth et al., 2006*). Nonmuscle myosin-2s are essential for the structural integrity of the cytoplasmic architecture during cell shape remodeling and motile events of eukaryotic cells,

*Figure 1 continued on next page*

*Figure 1 continued*

whereas all other myosins-2 play eminent roles in the contraction of smooth, cardiac and striated muscle cells (**Sellers, 2000**). Abbreviations used: NM2A: nonmuscle myosin-2A, NM2B: nonmuscle myosin-2B; NM2C: nonmuscle myosin-2C; SM: smooth muscle myosin-2; CardA: α-cardiac myosin-2; CardB: β-cardiac myosin-2; EO2: extraocular myosin-2; EMB: embryonic myosin-2; PERI: perinatal myosin-2; IIb: fast skeletal muscle myosin-2; IIx/d: skeletal muscle myosin-2; IIa: slow skeletal muscle myosin-2. (B) Architecture of the crystallized NM2C construct in the pre-powerstroke state. The myosin motor domain and the α-actinin repeats are shown in cartoon representation in green and grey color. The nucleotide is shown in spheres representation. *Inset,* Conserved key residues that interact with the nucleotide in the NM2C active site. The $F_o$-$F_c$ omit map of $Mg^{2+}$·ADP·$VO_4$ is contoured at 4σ. The salt bridge between switch-1 R261 and switch-2 E483 is highlighted. (C) Subdomain architecture of NM2C. The U50 kDa is shown in green, the L50 kDa in purple, the converter in grey, and the Nter in blue. The region shown in orange corresponds to the active site and the junction of U50 kDa and L50 kDa. The bound nucleotide is shown in spheres representation. The location of the SH1-SH2 helix and the relay helix in the L50 kDa is highlighted.

DOI: https://doi.org/10.7554/eLife.32742.002

The following figure supplement is available for figure 1:

**Figure supplement 1.** Myosin-2 ATPase cycle, expression constructs, and structural alignment of the NM2C $C_\alpha$ coordinates.

DOI: https://doi.org/10.7554/eLife.32742.003

F-actin has been determined (**von der Ecken et al., 2016**). The structure of the actomyosin complex, together with a detailed kinetic characterization of the motor properties, qualifies nonmuscle myosin-2C as the ideal human myosin to decipher the structure-function relationships in the myosin motor domain that underlie its kinetic signatures (**Heissler and Manstein, 2011**; **von der Ecken et al., 2016**).

To understand how structural adaptations lead to characteristic kinetic features, we solved the 2.25 Å crystal structure of the human nonmuscle myosin-2C motor domain (NM2C) (**Figure 1B**, **Table 1**). The NM2C pre-powerstroke state structure (ADP·$VO_4$) suggests that structural fine-tuning at the active site and a reduced interdomain connectivity in the motor domain define its kinetic signatures. Comparative structural analysis, ensemble solution kinetic studies, and in silico molecular dynamics (MD) simulations collectively demonstrate a communication pathway that connects the active site and the distal end of the motor domain. Disruption of the pathway uncouples the myosin ATPase activity from actin-activation, alters the ATP/ADP sensitivity and results in the loss of nonmuscle myosin-2 kinetic signatures. The allosteric coupling pathway may be of significance in intermolecular gating and load-sensitivity of nonmuscle myosins-2 that is important for their physiological function as cytoskeletal strain-sensor in the context of stress fibers, actin arcs, and cortical actin-based structures.

## Results

### Overall topology of pre-powerstroke NM2C

We determined a 2.25 Å pre-powerstroke state structure of human NM2C, one of the slowest myosins-2 characterized. Crystallization in the presence of the ATP analog ADP·$VO_4$ was achieved after truncation of the flexible N-terminal 45 residues extension and the fusion of the NM2C C-terminus to *Dictyostelium* α-actinin tandem repeats 1 and 2 (**Figure 1B**, **Figure 1—figure supplement 1B**, **Table 1**). This approach has been used successfully in prior structural and kinetic studies on NM2C and allows the detailed analysis of structure-function relationships in the myosin motor domain (**Heissler and Manstein, 2011**; **von der Ecken et al., 2016**).

The overall topology of NM2C shares the structural four-domain architecture of myosin-2 motor domains comprising the N-terminal subdomain (Nter), the upper 50 kDa subdomain (U50 kDa), the lower 50 kDa subdomain (L50 kDa), and the converter that terminates in the lever (**Figure 1C**) (**Rayment et al., 1993**). The nucleotide-binding active site is formed by structural elements of the U50 kDa and allosterically communicates with the actin-binding interface that is formed by distant structural elements of the U50 kDa and the L50 kDa. The NM2C $C_\alpha$ atoms superimpose with a root mean square deviation (*r.m.s.d.*) of 0.57 Å, 0.78 Å, and 0.73 Å to the closely related motor domain

**Table 1.** Crystallographic data collection and refinement statistics for human NM2C.

| Parameter | Hs NM2C-ADP·VO$_4$ |
|---|---|
| *X-ray data reduction statistics* | |
| Space group | $P22_12_1$ |
| *Unit cell dimensions* | |
| a, b, c | 81.12 Å, 125.61 Å, 153.96 Å |
| α,β,γ | 90°, 90°, 90° |
| Resolution | 97.33 Å – 2.25 Å |
| Last shell | 2.35 Å – 2.25 Å |
| Total measurements | 1116790 |
| No. of unique reflections | 75357 |
| Last shell | 9069 |
| Wavelength | 0.91841 Å |
| $R_{merge}$ | 0.12 |
| Last shell | 0.23 |
| I/σ(I) | 13.6 |
| Last shell | 2.63 |
| Completeness | 0.998 |
| Last shell | 1.00 |
| Multiplicity | 14.82 |
| Last shell | 14.97 |
| *Refinement statistics* | |
| Resolution | 34.77 Å–2.25 Å |
| Last shell | 2.31 Å–2.25 Å |
| No. of reflections (working set) | 71459 |
| No. of reflections (test set) | 3786 |
| *R*-factor | 0.2210 |
| Last shell | 0.2910 |
| *R*-free | 0.2410 |
| Last shell | 0.2920 |
| *No. of non-hydrogen atoms* | |
| Macromolecules | 7697 |
| Ligands | 33 |
| solvent | 666 |
| *Average B-factor* | |
| Protein | 76.06 Å$^2$ |
| Ligands | 35.59 Å$^2$ |
| Solvent | 63.37 Å$^2$ |
| *RMS deviations from ideal values* | |
| Bond lengths | 0.01 Å |
| Bond angles | 1.16° |
| Ramachandran favored | 96.22% |
| Ramachandran allowed | 3.78% |
| Ramachandran outliers | 0 |

DOI: https://doi.org/10.7554/eLife.32742.004

structures from chicken smooth muscle myosin-2 (PDB entry 1BR2), scallop striated muscle myosin-2 (PDB entry 1QVI), and *Dictyostelium* nonmuscle myosin-2 (PDB entry 2XEL) (*Figure 1—figure supplement 1C*) in the pre-powerstroke state.

NM2C pre-powerstroke state characteristics include a closed conformation of the nucleotide-binding motifs switch-1 and −2 in the active site (*Figure 1B*). In this conformation, the characteristic salt bridge between switch-1 R261 and switch-2 E483 that is pivotal for the hydrolysis of ATP is formed (*Rayment et al., 1993*; *Furch et al., 1999*; *Reubold et al., 2003*). The coordination of the co-crystallized ADP·VO$_4$ is similar to other myosin-2 structures in the pre-powerstroke state. The relay helix is in a kinked conformation, the central seven-stranded β-sheet is untwisted, and the converter domain and the adjacent lever arm in the up-position (*Figure 1B*). NM2C switch-1 and lever dihedral φ, ψ angles are substantially changed when compared to the pre-powerstroke state structures of chicken smooth muscle myosin-2 (PDB entry 1BR2), indicating that small arrangements in the active site are coupled to conformational changes at the distal end of the motor domain (*Figure 1—figure supplement 1C–D*, *Supplementary file 1*). Concomitantly, the relative orientation of converter and lever arm deviates by ~8–10° when compared to the high-resolution pre-powerstroke state crystal structure of chicken smooth muscle myosin-2 (PDB entry 1BR2) and resembled the orientation of the respective regions of scallop striated muscle myosin-2 (PDB entries 1QVI, 1DFL) motor domain structures (*Figure 1—figure supplement 1D*). In summary, the overall topology of NM2C resembles pre-powerstroke states of other class-2 myosins with subtle changes in the active site and the converter/lever arm orientation that may account for its different kinetic and mechanical output.

## Unique structural rearrangements in the NM2C active site in the pre-powerstroke state

Switch-1, switch-2, P-loop, and the purine-binding A-loop are the prototypic nucleotide-binding motifs in the myosin active site that undergo conformational changes in response to nucleotide binding and release. The active site is flanked by loop-3 and a loop that connects helices J and K in the U50 kDa, hereafter referred to as JK-loop (*Figure 2A*). The JK-loop constitutes a major connection between switch-1 and the nucleotide is of great functional significance as a hotspot for human cardiomyopathy-causing mutations in cardiac myosin-2 and the site of exon 7 in *Drosophila* muscle myosin-2 (*Figure 2—figure supplement 1A*) (*Miller et al., 2007*; *Van Driest et al., 2004*; *Havndrup et al., 2003*).

A slight reduction in the JK-loop length in NM2C causes an 8.8 Å shift between switch-1 and the U50 kDa and abolishes the formation of a tight interaction network with residues of the active site found in other class-2 myosins (*Figure 2B,C*, *Figure 2—figure supplement 1A,B,C*). Comparative analysis of interactions between the JK-loop and the switch-1 region in NM2C and scallop striated muscle myosin-2 (PDB entry 1QVI) shows that in the latter, JK-loop N321 is in hydrogen bond interaction with switch-1 N238 and located at a distance of 4.6 Å to the hydroxyl group of the C2' of the ADP ribose. The connectivity between switch-1 and the nucleotide is further strengthened by a hydrogen bond between N237 and the ADP ribose. Moreover, A-loop residue R128 forms hydrogen bonds with the ADP adenosine (3.2 Å) and E184 (2.8 Å) of the P-loop in the active site of striated muscle myosin-2. Strikingly, NM2C lacks all interactions described for scallop striated muscle myosin-2 due to the replacement of R128 with Q150 and JK-loop shortening. Both structural alterations increase the distance to the adenosine in the active site to 5.8 Å and disrupt constrains between swich-1 and the JK-loop (*Figure 2B,C,E*). The shift also increases the volume and hence the accessibility of the NM2C active site when compared to the transition state structures of chicken smooth muscle myosin-2 (PDB entry 1BR2), *Dictyostelium* nonmuscle myosin-2 (PDB entry 2XEL), and scallop striated muscle myosin-2 (PDB entry 1DFL,1QVI) (*Figure 2A*, *Figure 2—figure supplement 1B,D,E*). Changes in the spheres that describe the active site volumes indicate a 4.3-fold increase from 697 Å$^3$ in the case of scallop striated muscle myosin-2 (PDB entry 1QVI) to 3054 Å$^3$ for NM2C (*Figure 2—figure supplement 1D,E*). Strikingly, the enlarged accessibility of the active site region observed in the NM2C pre-powerstroke state structure resembles the conformation of the active site found in the actin-bound NM2C rigor state (PDB entry 5J1H), indicating that F-actin binding does not induce major structural rearrangements in the vicinity of the active site (*Figure 2C*). Together, these features demonstrate significant differences in the interaction pattern and geometry of the NM2C active site compared to other myosins-2.

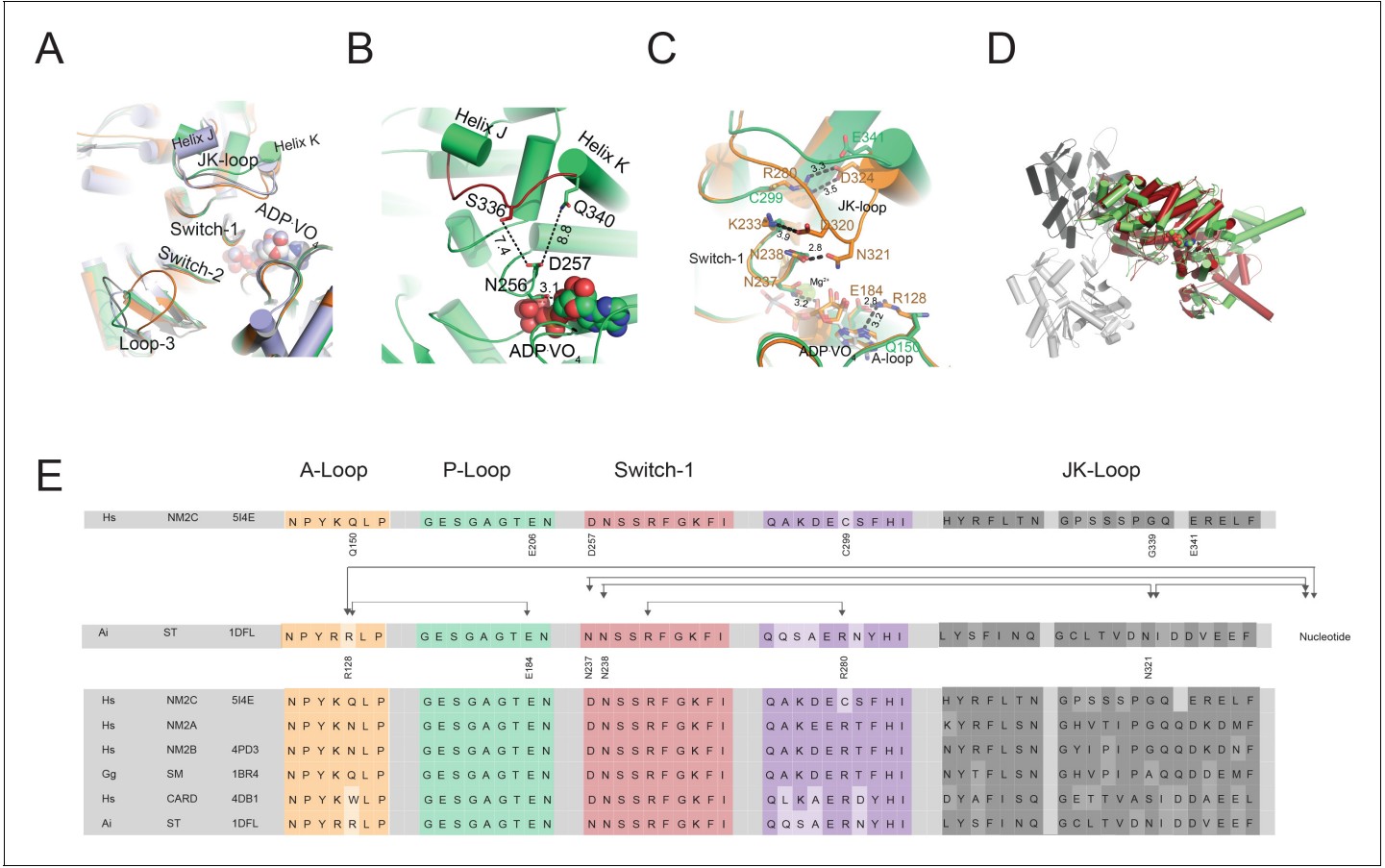

**Figure 2.** Conformational changes of the JK-loop in the myosin active site. (**A**) Top view on the NM2C active site in the pre-powerstroke state (green) superimposed on pre-powerstroke state structures from chicken smooth muscle myosin-2 (grey, PDB entry 1BR4), *Dictyostelium* nonmuscle myosin-2 (blue, PDB entry 2XEL), and scallop striated muscle myosin-2 (orange, PDB entry 1QVI). The ATP analog ADP·VO$_4$ is shown in spheres representation. (**B**) Conformation of the JK-loop in vicinity to the NM2C active site. The JK-loop flanks the active site and connects helices J and K. The distance between the residue Q340 of the JK-loop in the U50 kDa and the D257 of switch-1 of the active site is ~8.8 Å. The distance between residue S336 of the JK-loop and switch-1 D257 is ~7.4 Å. Switch-1 residue N256 interacts with α-phosphate (3.1 Å) and β-phosphate (3.5 Å) group of ADP·VO$_4$ in the active site. NM2C is colored in green/orange, the JK-loop is colored in brick red and ADP·VO$_4$ is shown in spheres. (**C**) Interactions between the JK-loop and the switch-1 region are compared between the NM2C (green) and scallop striated muscle myosin-2 (orange, PDB entry 1QVI). A-loop residue R128 is coordinating the interaction to the ADP adenosine in the active site of striated muscle myosin-2. The distance between the residues is 3.2 Å. R128 further forms a hydrogen bond (2.8 Å) with E184 of the P-loop. JK-loop N321 is in hydrogen bond interaction with switch-1 N238, located at a distance of 4.6 Å to the hydroxyl group of the C2' of the ADP ribose. The connectivity between switch-1 and the nucleotide is further strengthened by a hydrogen bond between N237 and the ADP ribose. NM2C lacks all interactions described for scallop striated muscle myosin-2 due to the replacement of R128 with Q150 and JK-loop shortening which increases the distance to the adenosine in the active site to 5.8 Å and disrupts constrains between swich-1 and the JK-loop. All residues in the JK-loop region are labeled for scallop striated muscle myosin-2 (PDB entry 1QVI). For NM2C only amino acid substitutions are labeled for legibility. (**D**) Superimposition of the NM2C pre-powerstroke state structure (green) and the actin-bound near-rigor actoNM2C complex (red) shows that the nucleotide-binding site does not undergo major structural changes. Actin subunits are colored in shades of grey and the nucleotide is shown in spheres representation. (**E**) Sequence alignment of select structural elements in the myosin motor domain that interact with the JK-loop. Interactions of A-loop R128 are highlighted with brackets for scallop striated muscle myosin-2 (PDB entry 1QVI). All highlighted interactions are absent in NM2C due to the presence of Q150 in the A-loop. Abbreviations used: *Hs* NM2C: human NM2C (NP_079005.3); *Hs* NM2A: human nonmuscle myosin-2A (NP_002464.1); NM2B: human nonmuscle myosin-2B (NP_005955.3); *Gg* SM: chicken smooth muscle myosin-2 (NP_990605.2); *Hs* CARD: human beta β-cardiac muscle myosin-2 (NP_000248.2); *Ai* ST: scallop striated muscle myosin-2 (P24733.1). PDB entries are indicated when available.

DOI: https://doi.org/10.7554/eLife.32742.005

The following figure supplement is available for figure 2:

**Figure supplement 1.** Active site characteristics in myosin-2 motor domains.
DOI: https://doi.org/10.7554/eLife.32742.006

The temperature factors obtained from the refined crystal structure show that the NM2C JK-loop is highly flexible despite its reduced length. We attribute the flexibility to the lack of constraints with switch-1 residues and the bound nucleotide (*Figure 2A,B,C,E*). Specifically, the connectivity of the JK-loop to switch-1 is reduced by the replacement of a conserved arginine (R280 in PDB entry 1QVI) with cysteine (C299) in the loop preceding helix I. This arginine further establishes an interaction with the highly conserved switch-1 K233 in scallop striated muscle myosin-2. The arginine to cysteine substitution in NM2C also abolishes the formation of a salt bridge with JK-loop residue E341 and hence disrupts the connection between the JK-loop and switch-1 (*Figure 2C,E*, *Figure 2—figure supplement 1C*). Consequently, the JK-loop loses its ability to sense conformational changes in the nucleotide binding motif switch-1 in response to ATP binding, hydrolysis, and product release.

The interaction between the JK-loop and the nucleotide is weakened by the replacement of an asparagine with G339, which abolishes hydrogen bond formation with switch-1 D257. Comparison with data from previous crystallographic studies shows that D257 replaces an asparagine (N238 in PDB entry 1QVI) in the switch-1 in striated muscle myosins-2. The asparagine interacts weakly with the ADP moiety in the pre-powerstroke state (*Figure 2B,D,E*, *Figure 2—figure supplement 1B,C*) and strongly in the near-rigor state (*Risal et al., 2004*; *Swank et al., 2006*). The nucleotide coordination in NM2C is further weakened by residue Q150 that replaces an arginine (R128 in PDB entry 1QVI) in the A-loop of scallop striated muscle myosin-2. The substitution disrupts coordinating interactions between the A-loop and the ADP adenosine (*Figure 2D*, *Figure 2—figure supplement 1B, C*). The replacement additionally impairs the formation of a salt bridge with the invariant E206 (E184 in PDB entry 1QVI) at the distal end of the P-loop that is predicted to be involved in nucleotide recruitment to the active site (*Figure 2D*, *Table 2*) (*Risal et al., 2004*). Further, the substitution of an arginine with K255 and a glutamate with H700 in NM2C is expected to weaken a tight and highly conserved interaction network between the active site and the Nter of NM2C (*Table 2*) (*Risal et al., 2004*). In summary, our comparative structural analysis of the active site of myosins-2 suggests that the interconnectivity of the highly conserved nucleotide switches and myosin subdomains is weaker in NM2C and likely other nonmuscle myosins-2 compared to fast sarcomeric myosins-2.

## Interdomain connectivity between converter, Nter, and lever

The interdomain connectivity between the converter and the Nter in the myosin-2 motor domain is established by the relay and the SH1-SH2 helix. The Nter controls the movement of the converter, which undergoes a large-scale rotation that drives the powerstroke during force generation (*Sasaki et al., 2003*; *Ramanath et al., 2011*; *Brenner et al., 2014*; *Llinas et al., 2015*; *Preller and Manstein, 2013*).

Residue R788 at the interface of converter, Nter, and lever is highly conserved in nonmuscle and smooth muscle myosins-2 and the only connecting hub between structural elements of the L50 kDa and the Nter in NM2C (*Figure 3A,B*). Notably, replacement of R788 with a lysine in muscle myosins-2 weakens the interaction between the aforementioned structural elements in all states of the myosin and actomyosin kinetic cycle (*Figure 3—figure supplement 1A,B*). In NM2C, R788 forms three main chain and five side chain interactions with residues of the SH1-SH2 helix, the converter, and the lever in the pre-powerstroke state in addition to a side chain interaction with a water molecule (2.8 Å). Specifically, the guanidinium group of the R788 side chain of NM2C forms hydrogen bonds with the main chain carbonyls of residues Q730 and G731 of the SH1 helix. The hydrophobic methylene groups of R788 are stabilized by SH1 helix F732 and the main chain oxygen and hydroxyl groups of N776 of the converter. Y518 from the relay helix interacts with residues G731 and F732 of the converter, thereby interlinking R788 with both structural elements. The side chain of R788 is in van der Waals distance (4.8 Å) from W525 of the relay loop. The main chain carbonyl of R788 further interacts with V791 at the converter/lever junction (*Figure 3A,B*). This interaction is important for the stabilization of the converter fold and the interface with the lever in the absence of F-actin. Notably, the R788 side chain:main chain interaction is also evident in crystal structures of nonmuscle myosins-2 in the nucleotide-free (PDB entry 4PD3) and the phosphate-release state (PDB entry 4PJK) (not shown).

We therefore hypothesize that (i) the tight interaction network formed by R788 is required for the precise positioning and coupling of the converter, the SH1-SH2 helix, the relay helix, and the lever arm throughout all steps of the myosin ATPase cycle (*Figure 1—figure supplement 1A*), and that (ii)

**Table 2.** Structure function relationships in the myosin-2 motor domain.

Interactions between residues in the active site involved in nucleotide binding and release kinetics based on this work and previous biochemical and structural studies on myosin motor domains (*Miller et al., 2007*; *Risal et al., 2004*; *Swank et al., 2006*; *Grammer et al., 1993*; *Szilagyi et al., 1979*). It is of note that myosin is a highly allosteric enzyme and nucleotide binding and release kinetic involve numerous interactions and subtle structural rearrangements of residues from different motor subdomains. Kinetic parameters from monomeric myosin motor domain constructs that are associated with a structural interaction are listed for direct comparison. An emerging trend from this analysis is that the myosin-2 kinetic cycle does not have a selectivity of ATP versus ADP. The presence of F-actin results in different allosteric communication pathways in myosins-2 and establishes ATP/ADP binding selectivity. Overall, nucleotide-binding rates are decreased for the group of nonmuscle and smooth muscle myosins-2 compared to myosins-2 from cardiac and skeletal muscle. The lacking salt bridge interactions between JK-loop, U50 kDa and switch-1 in nonmuscle myosins-2 results in decreased second-order binding rate constants for ATP ($K_1k_{+2}$) and ADP ($k_{+D}$) (*Figure 1—figure supplement 1A*). Either a salt bridge interaction or hydrophobic interactions between the A-loop and the P-loop of muscle and cardiac myosins-2 at the active site favor fast nucleotide binding kinetics and does not or only marginally discriminate between ADP and ATP. The lack of a salt bridge interactions can have different effects dependent on the coordinating residue in the A-loop: An asparagine in the A-loop of nonmuscle myosins-2A and −2B favors ADP over ATP binding to actomyosin. A glutamine in the NM2C A-loop, which has a longer side chain than asparagine, abolishes ATP/ADP sensitivity in NM2C and the closely related smooth muscle myosin-2. The number of salt bridge interactions between P-loop, switch-1, and the Nter correlates with the thermodynamic and kinetic coupling, and the actin-activated ADP release rates in all myosins-2. Abbreviations used: NM2A: human nonmuscle myosin-2A; human NM2B: nonmuscle myosin-2B (PDB entry 4PD3); NM2C: human nonmuscle myosin-2C; SM: chicken smooth muscle myosin-2 (PDB entry 1BR2); CARD: human β-cardiac myosin-2 (PDB entry 4DB1); *Oc* ST: rabbit striated muscle myosin-2 (PDB entry 1DFL).

| Myosin | Residue | Residue | Residue | Residue | Parameter | |
|---|---|---|---|---|---|---|
| | *JK-loop* | *U50 kDa* | *JK-loop* | *Switch-1* | *$K_1k_{+2}$* | *$K_{+D}$* |
| *Hs* NM2A (*Kovács et al., 2003*) | D315 | R272 | G312 | D230 | 0.56 | 0.55 |
| *Hs* NM2B (*Wang et al., 2003*) | D322 | R279 | G319 | D237 | 0.65 | 0.81 |
| *Hs* NM2C | E341 | C299 | G339 | D257 | 0.48 | 0.39 |
| *Gg* SM (*Cremo and Geeves, 1998*) | D328 | R285 | A325 | D243 | 2.1 | 1.8 |
| *Hs* CARD (*Deacon et al., 2012*) | D325 | R281 | S322 | D239 | 1.5 | 1.5 |
| *Oc* ST (*Ritchie et al., 1993*; *Kurzawa-Goertz et al., 1998*) | D323 | R280 | N321 | N238 | 3.9 | 1.7 |

| | Residue | Residue | Parameter | | | | |
|---|---|---|---|---|---|---|---|
| | *A-loop* | *P-loop* | *$k_{+AD}$* | *$K_1k_{+2}$* | *$k_{+D}$* | *$K_1k_{+2}$* | *$k_{+AD}$/ $K_1k_{+2}$* |
| *Hs* NM2A (*Kovács et al., 2003*) | N126 | E182 | 2.72 | 0.14 | 0.55 | 0.56 | 19.4 |
| *Hs* NM2B (*Wang et al., 2003*) | N130 | E186 | 2.41 | 0.24 | 0.81 | 0.65 | 10 |
| *Hs* NM2C | Q150 | E206 | 2.54 | 1.86 | 0.39 | 0.48 | 1.5 |
| *Gg* SM (*Cremo and Geeves, 1998*) | Q129 | E185 | 3.6 | 2 | 1.8 | 2.1 | 4.4 |
| *Hs* CARD (*Deacon et al., 2012*) | W130 | V186 | 4.4 | 1.1 | 1.5 | 1.5 | 1.6 |
| *Oc* ST (*Ritchie et al., 1993*; *Kurzawa-Goertz et al., 1998*) | R128 | E184 | 1.6 | 2.5 | 1.7 | 3.9 | 1.6 |

| | Residue | Residue | Residue | Parameter | | |
|---|---|---|---|---|---|---|
| | *P-loop* | *Switch-1* | *Nter* | *$K_{AD}/K_D$* | *$k_{-AD}/k_{-D}$* | *$k_{-AD}$* |
| *Hs* NM2A (*Kovács et al., 2003*) | E175 | K228 | H676 | 0.7 | 2.8 | 1.72 |
| *Hs* NM2B (*Wang et al., 2003*) | E179 | K235 | H683 | 0.2 | 0.7 | 0.35 |
| *Hs* NM2C | E199 | K255 | H700 | 0.11 | 1 | 0.65 |
| *Gg* SM (*Cremo and Geeves, 1998*) | E178 | K241 | H689 | 4.2 | 12 | 15 |
| *Hs* CARD (*Deacon et al., 2012*) | E179 | R237 | E677 | 42 | 103.3 | 150 |
| *Oc* ST (*Ritchie et al., 1993*; *Kurzawa-Goertz et al., 1998*) | E177 | R236 | E675 | 49 | 250 | 500 |

DOI: https://doi.org/10.7554/eLife.32742.007

different communication pathways between the active site and the distal end of the motor domain are employed by NM2C in the presence and absence of F-actin.

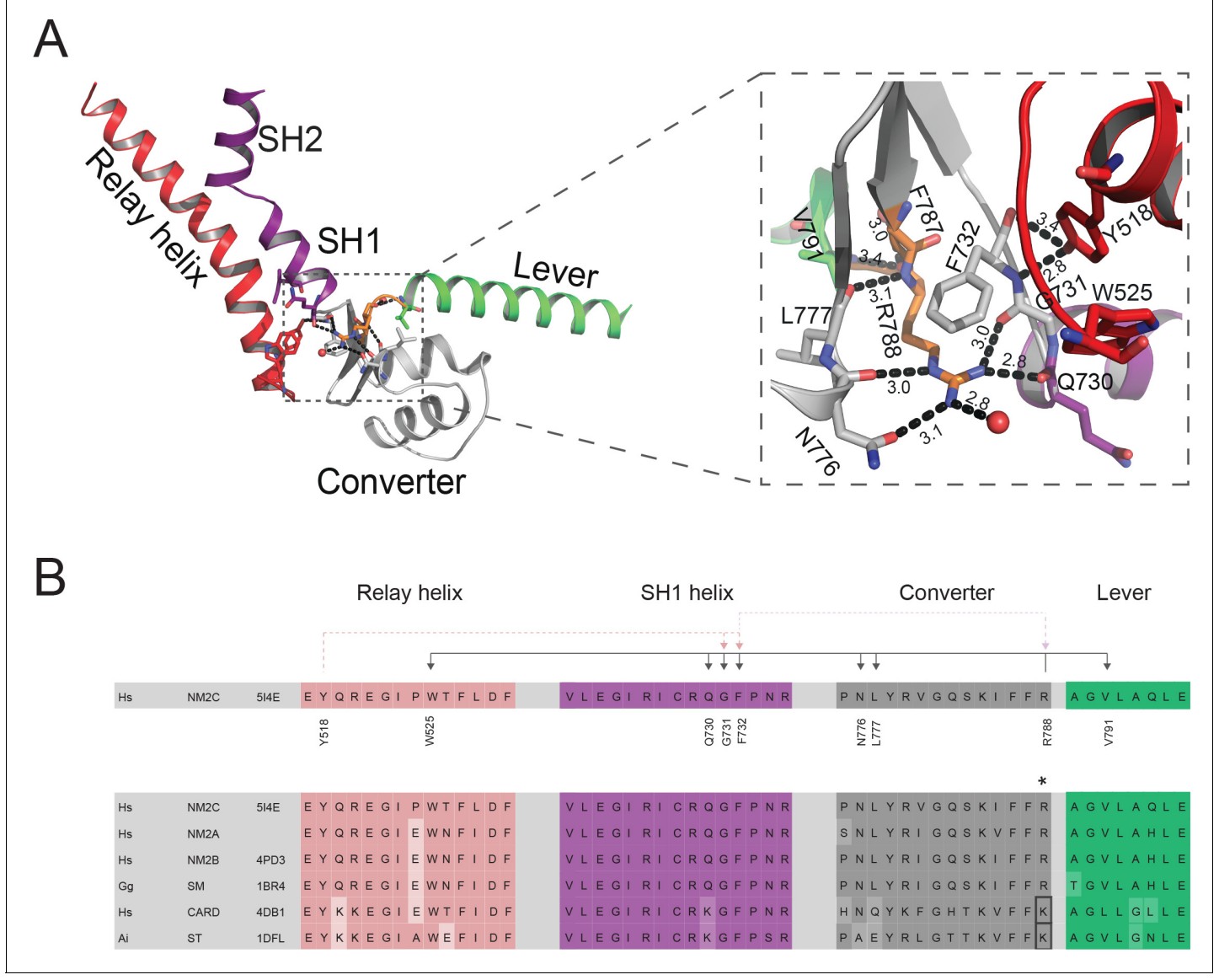

**Figure 3.** Interdomain connectivity at the converter/Nter/lever junction. (**A**) Interaction profile of R788 in the pre-powerstroke state. R788 is shown in orange colored sticks and the converter is colored in white, the relay helix in red, the SH1-SH2 helix in purple, and the lever arm in green colored cartoon representation. The inset shows a close-up view of the complete R788 interaction profile and is rotated 137° respective to the main panel. The guanidinium group of R788 forms hydrogen bonds (2.8 Å) with backbone oxygen atom of Q730 from the SH1 helix and the backbone oxygen atom of G731 (3.0 Å) of the converter. The δ-nitrogen atom of R788 interacts (3.0 Å) with N776 backbone oxygen atom of N776. The R788 guanidinium group interacts (3.1 Å) with the hydroxyl group of N776 of the converter. The backbone nitrogen atom of R788 interacts (3.1 Å) with the carbonyl group of L777 of the converter. The backbone carbonyl group of R788 interacts (3.4 Å) with the backbone nitrogen of V791 of the lever as well as a water molecule (3.0 Å). The hydroxyl group from relay helix Y518 interacts with the backbone nitrogen atom of G731 (2.8 Å) and backbone oxygen atom of F732 (3.4 Å). F732 forms hydrophobic interactions with the methylene groups of R788 with the latter positioned in van der Waals distance to relay loop W525. All the amino acids involved in interactions with R788 are shown as sticks and water molecules as spheres. (**B**) Sequence alignment of selected regions from relay helix, SH1 helix, converter and lever arm shows the high sequence conservation within the myosin-2 motor domain. The asterisk indicates the invariant, positively charged residue corresponding to NM2C R788. The interactions of NM2C R788 with structural elements of the L50 kDa, the converter, and the lever are highlighted. Abbreviations used: *Hs* NM2C: human NM2C (NP_079005.3); *Hs* NM2A: human nonmuscle myosin-2A (NP_002464.1); NM2B: human nonmuscle myosin-2B (NP_005955.3); *Gg* SM: chicken smooth muscle myosin-2 (NP_990605.2); *Hs* CARD: human beta β-cardiac muscle myosin-2 (NP_000248.2); *Ai* ST: scallop striated muscle myosin-2 (P24733.1). PDB entries are indicated when available. Lysine residues that replace R788 in cardiac and striated muscle myosins-2 highlighted in the boxed area.

DOI: https://doi.org/10.7554/eLife.32742.008

The following figure supplement is available for figure 3:

**Figure supplement 1.** Interdomain connectivity at the converter/Nter/lever junction in muscle myosins-2.

*Figure 3 continued on next page*

*Figure 3 continued*

DOI: https://doi.org/10.7554/eLife.32742.009

## Kinetic consequences after the disruption of the converter/Nter/lever interface

To experimentally probe for a possible effect of R788 on myosin motor function, we performed comparative ensemble solution kinetic studies with NM2C and two mutants in which the interaction between residue 788 and interacting residues at the converter/Nter/lever interface is weakened. In the conservative R788K mutant, a lysine, as it is found in sarcomeric myosins-2 replaces the converter R788. In R788E, a glutamate replaces the converter R788. Rationale for the design of the conservative R788K mutant was to reduce the number of electrostatic side chain:side chain and side chain: main chain interactions compared to NM2C. Based on the NM2C pre-powerstroke state structure, the side chain of K788 interacts with the main chain carbonyl of residue G731 (3.1 Å) of the SH1 helix. The main chain carbonyl of K788 is further predicted to interact with the main chain of V791 at the converter/lever junction. Other interactions with elements of the SH1-SH2 helix, the converter, the lever, and the water molecule (5 Å) as observed in NM2C are lost (*Figure 3—figure supplement 1C*). The charge reversal mutant R788E was designed to further reduce the number of side chain: side chain and side chain:main chain interactions. Interactions between the side chain carboxyl group of E788 and other residues at the converter/Nter/lever interface are abolished. The distance to the water molecule increases to 4.9–6.3 Å and does not allow the formation of a hydrogen bond. The main chain:main chain interaction with V791 is preserved and represents the only connection with the converter/lever junction (Figure 3—figure supplement 1D).

Confirming our hypotheses, transient kinetic changes in R788E are more severe than in R788K and mainly affect nucleotide binding and release kinetics (*Table 3*). Transient kinetic changes are more pronounced in the absence of F-actin, as seen in single-turnover measurements (*Figure 4A,B*). Most importantly, nonmuscle myosin-2 specific transient kinetic signatures such as a high $k_{+AD}$/$K_1k_{+2}$ ratio ($k_{+AD}/K_1k_{+2} \sim 2$–$20$) are absent in R788E ($k_{+AD}/K_1k_{+2} \sim 0.02$) (*Table 3*, *Figure 4—figure supplement 1A–D*) (*Heissler and Manstein, 2011*; *Kovács et al., 2003*; *Wang et al., 2003*). This feature is also described for conventional myosins-2 from cardiac and striated muscle that bind ATP and ADP with similar rates (*Table 4*) (*Marston and Taylor, 1980*). Actin-activation of the ADP release ($k_{-AD} = 0.68 \pm 0.01$ s$^{-1}$ for NM2C, $k_{-AD} = 0.59 \pm 0.01$ s$^{-1}$ for R788K, and $k_{-AD} = 0.19 \pm 0.01$ s$^{-1}$ for R788E) results in a kinetic coupling constant $k_{-AD}/k_{-D}$ of $\sim 3.8$ for R788E, whereas neutral or negative kinetic coupling are features of NM2C ($k_{-AD}/k_{-D} \sim 0.7$), R788K ($k_{-AD}/k_{-D} \sim 1.1$), and other human nonmuscle myosins-2 (*Table 4*) (*Heissler and Manstein, 2011*; *Kovács et al., 2003*; *Wang et al., 2003*). The changes in ADP binding and release kinetics of R788E result in a 10-fold increase in the ADP dissociation equilibrium constant $K_{AD}$ ($K_{AD} \sim 0.26$ µM for NM2C and $\sim 3$ µM for R788E). The thermodynamic coupling ($K_{AD}/K_D = 5$) for R788E is 42-times stronger than for NM2C (*Table 3*). R788E displays an extraordinary slow ADP binding rate constant ($k_{+AD} = 0.03 \pm 0.001$ µM$^{-1}$s$^{-1}$) (*Table 3*, *Figure 4—figure supplement 1D*) in the presence of F-actin, whereas the kinetic constants for the interaction between actomyosin and ATP, $K_1k_{+2}$, $k_{+2}$, and $1/K_1$ are only marginally affected when compared to NM2C (2). The F-actin affinity in the absence and presence of ADP ($K_A$, $K_{DA}$) of NM2C and R788E is similar (*Table 3*), whereas $K_{DA}$ is three- to five-fold higher in R788K (*Table 3*). F-actin can activate the steady-state ATPase activity of R788E to approximately half the $k_{cat}$ of NM2C ($k_{cat} = 0.2 \pm 0.01$ s$^{-1}$ for R788E and $k_{cat} = 0.37 \pm 0.02$ s$^{-1}$ of NM2C) (*Table 3*) under steady-state conditions. The steady-state parameter $k_{cat} = 0.26 \pm 0.2$ s$^{-1}$ for R788K is in between the respective parameters for NM2C and R788K. The duty ratio at an F-actin concentration of 190 µM and saturating [ATP] increases from $\sim 0.3$ for NM2C and R788K to $\sim 1$ for R788E due to the decreased $k_{cat}$ and the decreased actin-activated ADP release rate $k_{-AD}$. This feature makes $k_{-AD}$ likely to rate-limit the kinetic cycle of R788E, whereas the actin-activated P$_i$ release is expected to rate-limit the kinetic cycles of NM2C, R788K, and other myosins-2 (*Heissler and Sellers, 2016*; *Heissler and Manstein, 2011*; *Kovács et al., 2003*; *Wang et al., 2003*; *Woodward et al., 1995*; *Ritchie et al., 1993*).

The slow nucleotide binding and release kinetics of R788E suggest that the R788-mediated interaction at the converter/Nter/lever interface is allosterically communicated to the active site. This is in

**Table 3.** Steady-state and transient state kinetic parameters of NM2C, R788K, and R788E.

Numbering of the kinetic constants refers to *Figure 1—figure supplement 1A*. Measurements of transient kinetic parameters that rely on a change in intrinsic tryptophan fluorescence are not experimentally accessible for R788E. Normal and bold face notation denote the respective kinetic constants in the absence and presence of F-actin.

| Parameter | Signal or calculation | NM2C | R788K | R788E |
|---|---|---|---|---|
| *Steady-state ATPase* | | | | |
| $k_{cat}$ (s$^{-1}$) | NADH assay* | 0.37 ± 0.02 | 0.26 ± 0.02 | 0.2 ± 0.01 |
| $K_{app}$ (µM) | NADH assay* | 129.4 ± 17.2 | 110.5 ± 15.89 | 45.9 ± 13 |
| $k_{cat}/K_{app}$ (µM$^{-1}$s$^{-1}$) | NADH assay | ~0.003 | ~0.002 | ~0.004 |
| *ATP interaction* | | | | |
| $K_1 k_{+2}$ (µM$^{-1}$s$^{-1}$) | Tryptophan | 0.39 ± 0.01 | 0.18 ± 0.01 | Not accessible[¶] |
| $K_1 k_{+2}$ (µM$^{-1}$s$^{-1}$) | d-mantATP | 0.48 ± 0.01 | 0.22 ± 0.01 | 0.09 ± 0.005 |
| $1/K_{0.5}$ (µM) | Tryptophan | 46.23 ± 5.22 | 158.03 ± 9 | Not accessible[¶] |
| $k_3 + k_{-3}$ (s$^{-1}$) | Tryptophan | 24.2 ± 0.86 | 37.31 ± 0.55 | Not accessible[¶] |
| $\boldsymbol{K_1 k_{+2}}$ (µM$^{-1}$s$^{-1}$) | Pyrene-actin | 1.86 ± 0.03 | 1.03 ± 0.03 | 2.11 ± 0.05 |
| $\boldsymbol{K_1 k_{+2}}$ (µM$^{-1}$s$^{-1}$) | d-mantATP | 1.64 ± 0.04 | 0.73 ± 0.02 | 1.37 ± 0.04 |
| $\boldsymbol{1/K_1}$ (µM) | Pyrene-actin | ~318 | ~826 | ~380 |
| $\boldsymbol{k_{+2}}$ (s$^{-1}$) | Pyrene-actin | 643.06 ± 22.6 | 767.43 ± 21.27 | 579.27 ± 12.81 |
| *ADP interaction* | | | | |
| $k_{+D}$ (µM$^{-1}$s$^{-1}$) | d-mantADP | 0.39 ± 0.01 | 0.21 ± 0.02 | 0.12 ± 0.01 |
| $k_{-D}$ (s$^{-1}$) | d-mantADP[†] | 0.94 ± 0.07 | 0.58 ± 0.05 | 0.06 ± 0.01 |
| $k_{-D}$ (s$^{-1}$) | d-mantADP[‡] | 0.39 ± 0.01 | 0.51 ± 0.01 | 0.05 ± 0.001 |
| $k_{-D}$ (s$^{-1}$) | Tryptophan[‡] | 0.47 ± 0.02 | 0.24 ± 0.001 | Not accessible[¶] |
| $K_D$ (µM) | $k_{-D}/k_{+D}$ | ~1–2.4 | ~1.1–2.8 | ~0.5 |
| $\boldsymbol{k_{+AD}}$ (µM$^{-1}$s$^{-1}$) | d-mantADP | 2.54 ± 0.18 | not accessible** | 0.03 ± 0.001 |
| $\boldsymbol{k_{-AD}}$ (s$^{-1}$) | d-mantADP[b] | 0.65 ± 0.06 | not accessible** | 0.09 ± 0.03 |
| $\boldsymbol{k_{-AD}}$ (s$^{-1}$) | Light scattering[‡] | 0.39 ± 0.01 | 0.42 ± 0.002 | 0.15 ± 0.01 |
| $\boldsymbol{k_{-AD}}$ (s$^{-1}$) | d-mantADP[‡] | 0.68 ± 0.01 | 0.59 ± 0.01 | 0.19 ± 0.01 |
| $\boldsymbol{k_{-AD}}$ (s$^{-1}$) | Pyrene-actin[‡] | 0.48 ± 0.01 | 0.46 ± 0.01 | 0.14 ± 0.01 |
| $\boldsymbol{K_{AD}}$ (µM) | $\boldsymbol{k_{-AD}/k_{+AD}}$ | ~0.29 | Not accessible | ~2.68 |
| *Actin interaction* | | | | |
| $k_{+A}$ (µM$^{-1}$s$^{-1}$) | Light scattering[§,#] | 2.49 ± 0.07 | 1.65 ± 0.03 | 0.66 ± 0.03 |
| $k_{-A}$ (s$^{-1}$) | Pyrene-actin[§,#] | ~0.15 | ~0.04–0.07 | ~0.031 |
| $K_A$ (nM) | $k_{-A}/k_{+A}$ | ~60 | ~24–42 | ~47 |
| $k_{+DA}$ (µM$^{-1}$s$^{-1}$) | Light scattering[§] | 0.53 ± 0.01 | 2.53 ± 0.05 | 0.21 ± 0.01 |
| $k_{-DA}$ (s$^{-1}$) | Pyrene-actin[§] | ~0.004 | ~0.067–0.11 | ~0.002 |
| $K_{DA}$ (nM) | $k_{-DA}/k_{+DA}$ | ~8 | ~26–43 | ~9 |

*In 50 mM NaCl, 2 mM MgATP, T = 25°C.

[†]From y-intercept.

[‡]From chasing experiment.

[§]Measured in salt free stopped-flow buffer.

[#]Samples were treated with apyrase prior to the assay to ensure rigor conditions.

[¶]The lack of a change in intrinsic fluorescence upon nucleotide binding to R788E precludes the direct measurement of this kinetic parameter.

**Small amplitudes prevent the direct measurement of this parameter.

DOI: https://doi.org/10.7554/eLife.32742.012

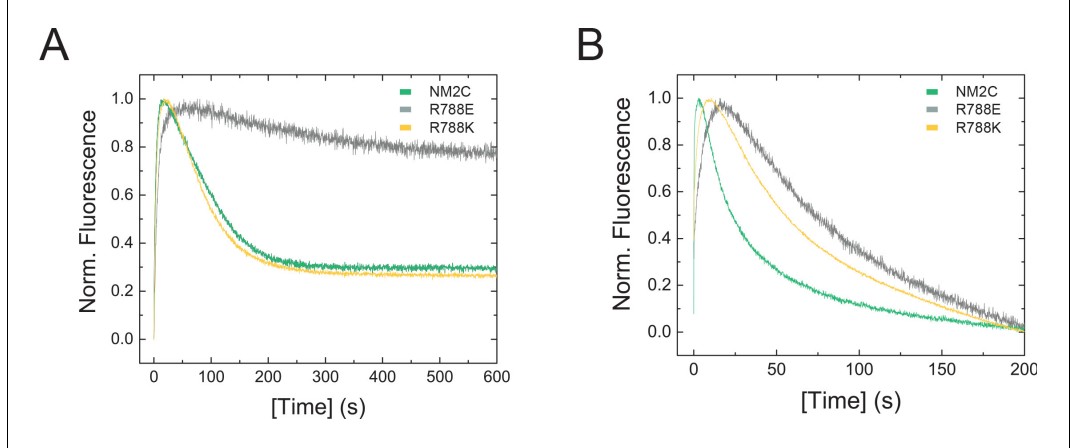

**Figure 4.** Transient kinetic features of NM2C and R788E. (**A**) Interaction between NM2C/R788K/R788E with ATP under single-turnover conditions. Binding 0.375 μM d-mantATP to 0.5 μM myosin results in a transient fluorescence increase that is followed by a short plateau (hydrolysis) and a slow decrease in mantADP fluorescence that is associated with its release. All three phases are reduced in R788E (grey) compared to NM2C (green) and R788K (yellow). The very slow decrease of the fluorescence signal in R788E indicates that either the ATP hydrolysis rate or a subsequent release rate of the hydrolysis products are severely decreased when compared to NM2C and R788K. (**B**) Interaction between 0.25 μM pyrene-actoNM2C/R788K/R788E with 0.15 μM ATP under single-turnover conditions. Color coding is according to (**A**).

DOI: https://doi.org/10.7554/eLife.32742.010

The following figure supplement is available for figure 4:

**Figure supplement 1.** Nucleotide binding characteristics of NM2C, R788K, and R788E and disease causing NM2C mutations.

DOI: https://doi.org/10.7554/eLife.32742.011

line with the observation that R788E does not show a nucleotide-induced change in the intrinsic tryptophan fluorescence signal during steady-state fluorescence measurements and transient-state kinetic assays (*Figure 5D,F*, *Figure 4—figure supplement 1E*, *Table 3*). The fluorescence change observed with NM2C is attributed to a conformation-induced change in the microenvironment of the conserved relay loop W525, which is in van der Waals distance to R788. W525 is generally

**Table 4.** Comparative analysis of kinetic signatures of monomeric myosin-2 motor domain constructs.
Abbreviations used: NM2A: human nonmuscle myosin-2A, NM2B: human nonmuscle myosin-2B (PDB entry 4PD3); NM2C: human nonmuscle myosin-2C; SM: chicken smooth muscle myosin-2 (PDB entry 1BR2); CARD: human/chicken β-cardiac myosin-2 (PDB entry 4DB1); ST: rabbit/scallop striated muscle myosin-2 (PDB entry 1DFL).

| Myosin | $k_{cat}/K_{app}$ | $k_{+AD}/K_1k_{+2}$ | $K_{AD}/K_D$ | $k_{-AD}/k_{-D}$ | $K_{DA}/K_A$ | Duty ratio |
|---|---|---|---|---|---|---|
| *Hs* NM2C | 0.003 | 1.5 | 0.11 | 1 | 0.13 | ~0.3[‡] |
| *Hs* R788K | 0.002 | n.d. | n.d. | 1 | 1 | ~0.3[‡] |
| *Hs* R788E | 0.004 | 0.26 | 5 | 3 | 0.19 | ~1[$] |
| *Hs* NM2A (*Kovács et al., 2003*) | 0.002 | 19.4 | 0.7 | 2.8 | 2 | 0.1 |
| *Hs* NM2B (*Wang et al., 2003*) | 0.002 | 10 | 0.2 | 0.7 | 0.4 | 0.37 |
| *Gg* SM (*Marston and Taylor, 1980*; *Cremo and Geeves, 1998*) | 0.07 | 4.4 | 4.2 | 12 | 6.9 | <0.05 |
| *Hs/Gg* CARD (*Marston and Taylor, 1980*; *Deacon et al., 2012*) | 0.07[†] | 1.6 | 42 | 103.3 | 23.9 | <0.02 |
| *Oc/Ai* ST (*Ritchie et al., 1993*; *Kurzawa-Goertz et al., 1998*; *Wagner, 1981*; *Harris and Warshaw, 1993*) | 1.6* | 1.6* | 49 | 250* | 30* | <0.04 |

*Rabbit striated muscle myosin-2.

[†]Chicken cardiac muscle myosin-2.

[‡]Calculated at 190 μM F-actin and saturating [ATP].

DOI: https://doi.org/10.7554/eLife.32742.013

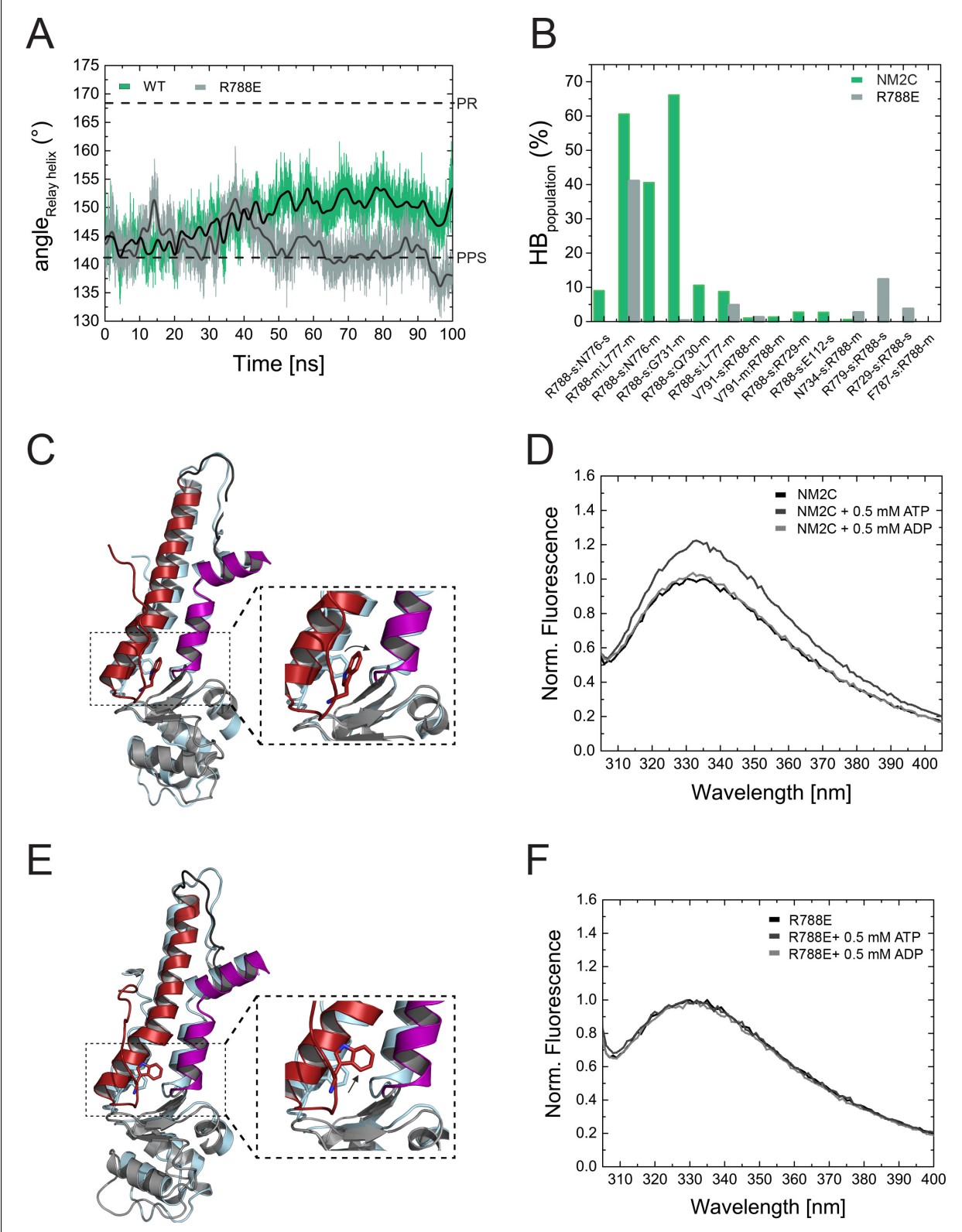

**Figure 5.** Structural importance of R788 at the converter/Nter/lever junction. (**A**) Relay helix angle as a function of MD simulation time, as monitored by the angle between $C_\alpha$ atoms of residues S489, M510, and E521 along the trajectories. The relay helix straightens in NM2C with a steady increase in the angle of the relay helix from approximately 145° to 150°, while the angle does not change significantly in R788E and fluctuates around 145° throughout the 100 ns time course of the simulation. Values for the relay helix angle observed in crystal structures of pre-power stroke (PPS) and post-rigor (PR) are

*Figure 5 continued on next page*

*Figure 5 continued*

indicated by dotted lines. (B) Population of hydrogen bonds (HB) between R788 (NM2C) or E788 (R788E) and surrounding structural elements over the simulation time of 100 ns. The abbreviations s and m indicate side chain and main chain. (C) Dynamics and conformational changes in NM2C during MD simulations. Snapshots from the start (0 ns simulation time) and end conformations (100 ns simulation time) are shown in light cyan and colored cartoon representation, respectively. The relay helix is shown in red, the SH1-SH2 helix in purple and the converter in grey. Relay loop W525 is shown in red in stick representation. The insets show a close-up view of the conformational changes of W525 along the simulation trajectory. (D) Tryptophan fluorescence emission spectrum of 4 µM NM2C in the absence of nucleotide or the presence of 0.5 mM ATP or 0.5 mM ADP. (E) Dynamics and conformational changes in R788E during MD simulations. Snapshots from the start (0 ns simulation time) and end conformations (100 ns simulation time) are shown in light cyan and colored cartoon representation, respectively. The relay helix is shown in red, the SH1-SH2 helix in purple and the converter in grey. Relay loop W525 is shown in red in stick representation. The insets show a close-up view of the conformational changes of W525 along the simulation trajectory. (F) Tryptophan fluorescence emission spectrum of 4 µM R788E in the absence of nucleotide or presence of 0.5 mM ATP or 0.5 mM ADP.

DOI: https://doi.org/10.7554/eLife.32742.014

The following figure supplement is available for figure 5:

**Figure supplement 1.** Stability of the salt bridge between switch-1 and switch-2 in MD simulations of NM2C, R788K, and R788E.

DOI: https://doi.org/10.7554/eLife.32742.015

regarded as a direct indicator for the switch-2 induced converter rotation (*Malnasi-Csizmadia et al., 2001*; *Batra and Manstein, 1999*).

Charge reversal mutations have been successfully used in in vitro and in vivo studies to probe for the influence of the converter/U50 kDa interface on myosin-2 performance (*Ramanath et al., 2011*; *Bloemink et al., 2016*; *Kronert et al., 2015*; *Kronert et al., 2014*). As expected, the structural consequences of the charge reversal mutation R788E are more severe than for the R788K mutant. R788K and R788E show prominent changes in virtually all parameters of the myosin and actomyosin ATPase cycle, again with more serious consequences occurring for R788E (*Table 3*). In summary, our steady-state and transient state kinetic analysis of NM2C, R788K, and R788E suggest that R788 is a key residue and is essential for allosteric communication pathway between the active site and the converter/Nter/lever interface.

## Allosteric communication pathway between the active site and the distal end of the motor domain

R788 is a hub amino acid and forms the center of a cluster of interactions that connect the converter, the SH1-SH2 helix, the relay helix, and the lever (*Figure 3A,B*). To understand its dynamic interactions that are allosterically communicated to the active site, we performed comparative MD simulations of NM2C, R788E, and R788K in explicit water with $Mg^{2+} \cdot$ATP bound to the active site over a time of 100 ns. The co-crystallized ADP·VO₄ was replaced by ATP since we found that the presence of ATP, while starting with the crystallized NM2C pre-power stroke structure, led to a straightening of the relay helix that connects the active site and the converter during our simulations. Likewise, the converter undergoes a 27° rotation that directs lever arm motion (*Figure 5A,C*). This straightening is solely caused by the exchange of the co-crystallized ADP·VO₄ to ATP, since neither the replacement with ADP nor the absence of nucleotide led to relay straightening, converter rotation, or larger changes in the active site in control simulations. Therefore, the use of ATP in our simulations allowed us to analyze the effect of the R788E and R788K mutations during the dynamic initial converter rotation and relay straightening.

As expected, the characteristic salt bridge between switch-1 R261 and switch-2 E483 of the active site that is critical for the hydrolysis of ATP is stable throughout the time course of the simulation for NM2C (*Figure 5—figure supplement 1A*). Negligible shifts of the active site P-loop and switch-1 are detectable during the simulations, which correlate with a minor shift of the ATP. The nucleotide coordination is however unchanged except for an interaction of switch-2 peptide backbone of G481 with the γ-phosphate of ATP (*Figure 5—figure supplement 1C,D*). This critical interaction, used to define the closed and open states of switch-2, is lost along the MD simulations (*Figure 5—figure supplement 1D*). This loss of interaction is due to a 2 to 2.5 Å *r.m.s.d.* rearrangement of switch-2. The cation $Mg^{2+}$ remains bound to the side chains of S260 and T205, as well as the α- and γ-phosphates of ATP within interaction distances around 2 Å. After 100 ns, the position of the converter and the adjacent lever show an orientation between the post-rigor and pre-powerstroke state, as

compared to the crystal structures of *Dictyostelium* nonmuscle myosin-2 motor domain (PDB entries 1FMW, 2XEL) (*Figure 5C*). This intermediate state resembles the proposed initial phase of the see-saw mechanism for the recovery stroke of myosin (*Koppole et al., 2007*). The see-saw mechanism proposes a two-phase swinging of the lever arm, with an initial ~25° rotation of the converter and lever arm, linked to the formation of a hydrogen bond between G457 and the γ-phosphate, and a second phase, featuring the remaining ~40° rotation and the complete closing of switch-2. Our simulations appear to show the initial phase with a 27° rotation of the converter and lever arm along the trajectories, an initial rearrangement of switch-2 and the associated loss of the critical interaction between G481 (corresponds to G457 in *Dictyostelium* myosin-2) and the γ-phosphate of ATP. The important interactions between G481, A480 and N499 in the relay helix, which are suggested to transmit the see-saw information to the relay helix, are stable throughout the simulations. With the movement of the relay helix, the indole ring of relay loop W525 changes its conformation by 70–80°, which agrees with the experimentally observed nucleotide-induced change in the intrinsic trypto-phan fluorescence signal in NM2C (*Figure 5B,C*, *Figure 4—figure supplement 1E*). The hub amino acid R788 is in transient interactions with main chain and side chain atoms of 10 amino acids of the SH1 helix and the converter during the 100 ns time course of the simulation (*Figure 5B*).

MD simulations for R788E indicate that all interdomain interactions between R788E of the converter and the SH1 helix are lost (*Figure 5B*). The straightening angle of the relay helix remains constant and abolishes a detectable converter rotation (*Figure 5A,E*). This allosteric decoupling at the distal end of the myosin motor domain is translated further upstream via the relay helix and results in a pronounced 6 Å conformational change of a loop that connects the γ-phosphate sensor switch-2 of the active site and the relay helix. As a consequence, the salt bridge between switch-2 E483 and switch-1 R261 appears less stable in the MD simulations of R788E as compared to NM2C (*Figure 5— figure supplement 1A,B*). The importance of the salt bridge between both switches for the ATP hydrolysis is established and expected to directly contribute to the experimentally observed impaired nucleotide binding, hydrolysis, and release kinetics of R788E in transient-state kinetic assays (*Table 3*, *Figure 4A*, *Figure 4—figure supplement 1A,D*) (*Furch et al., 1999*; *Friedman et al., 1998*; *Ruppel and Spudich, 1996*). Moreover, the side chain of switch-1 N256 changes its orientation, thereby impairing hydrogen bond interactions with the α- and β-phosphate moieties of the nucleotide and constrains of switch-1. The lack of coordinating interactions may be linked to the very slow nucleotide binding and release rate constants (*Figure 4A*, *Figure 4—figure supplement 1A*, *Table 3*).

The side chain of relay loop W525 does not undergo a conformational change throughout the time course of the simulation for R788E, supporting our experimental observation that mutant construct R788E does not exhibit a nucleotide-induced change in its intrinsic tryptophan fluorescence signal (*Figure 5D,F*, *Figure 4—figure supplement 1E*). The direct comparison of the dynamic inter-action and conformational signatures of W525 in NM2C and R788E reveals that R788 is in van der Waals distance (5.4 Å) from relay loop W525 at the start of the simulations. W525 transiently inter-acts with I523 (18%) and Q519 (14%), changes its conformation by 70–80° and increases the distance to NM2C R788 to 7.3 Å after 100 ns simulation time. In comparison, the distance between W525 and E788 increases to 9.9 Å during the simulation for R788E. The lack of a conformational change in W525 along the simulation trajectory results in transient interactions with Q515 (55%) and Q730 (14%) instead of the interactions with I532 and Q519 as seen in NM2C.

MD simulations for R788K indicate that K788 forms transient interactions with 10 main chain and side chain atoms of the SH1 helix and the converter (*Figure 5—figure supplement 1E*). The interac-tion patters are similar but not identical to NM2C (*Figure 5—figure supplement 1E*). These subtle changes are allosterically propagated from the converter/Nter/lever interface at the distal end of the myosin motor domain via the relay helix to switch-2 of the active site. The salt bridge between switch-2 E483 and switch-1 R261 is not significantly populated after 10 ns simulation time (*Figure 5— figure supplement 1F*) and hydrogen bond interactions between switch-1 N256 and the α- and β-phosphate moieties of the nucleotide in R788K break along the simulation trajectory. The dynamic features in the R788K active site resemble those observed for R788E (*Figure 5—figure supplement 1B,F*) and underline that the precise coordination of R788 at the interface of converter, the SH1-SH2 helix, the relay helix, and the lever is pivotal for the allosteric communication in the NM2C motor domain.

In summary, our in silico and experimental in vitro data show that R788 is a distant allosteric modulator of switch-2 dynamics at the active site that impacts nucleotide binding and release kinetics in the actin-detached states. Moreover, our data support a model of R788 as a quencher of W525 fluorescence (*Figure 5D,F*, *Figure 4—figure supplement 1E*).

## Discussion

The experiments presented here collectively demonstrate an allosteric communication pathway from the distal end of the myosin motor domain that, together with substitutions of several key residues in or in vicinity to the active site, account for NM2C-specific kinetic properties. Disruption of the pathway by mutation of R788 to a glutamate causes the loss of its enzymatic signatures and results in a high duty ratio motor. Importantly, several key residues involved in the allosteric communication pathway are implicated in the onset and progression of debilitating human diseases (*Fu et al., 2016*; *Donaudy et al., 2004*; *Kim et al., 2017*). Missense mutation G376C is in proximity to residues C324 of helix J and R328 of the JK-loop (*Figure 4—figure supplement 1F*). Based on its location, we suggest that this substitution may interfere with the nucleotide binding and release kinetics from the NM2C active site. Missense mutation R726S is in the SH1-SH2 helix and the guanidinium group of the wild-type arginine interacts (3.3 Å) with the NM2C Nter. The serine residue is expected to disrupt this interaction because of the shorter side chain and different charge when compared to arginine. It is therefore likely that these mutations may impact the interaction of the motor domain with nucleotides, thereby contributing to impaired tension-sensing and maintenance of nonmuscle myosin-2C in the human cochlea (*Fu et al., 2016*; *Donaudy et al., 2004*; *Kim et al., 2017*).

### R788 is part of a conserved pathway that connects the active site and the converter

R788 is part of an allosteric communication pathway that connects the converter at the distal end of the myosin motor domain via the relay helix with switch-2 of the active site (*Figure 5—figure supplement 1*). At the active site, the reduced number of hydrogen bond interactions between P-loop, switch-1, and the Nter together with the reduced interconnectivity within the motor domain contribute to the low thermodynamic ($K_{AD}/K_D$) and kinetic ($k_{AD}/k_{-D}$) coupling efficiency and the efficiency of F-actin to displace ADP and $P_i$ during the catalytic cycle in NM2C and likely other nonmuscle and smooth muscle myosins-2 compared to cardiac and striated muscle myosins-2 (*Table 2*) (*Risal et al., 2004*). In summary, the discussed structural details contribute to the kinetic features of NM2C that are overall characterized by a slow steady-state ATPase activity and higher duty ratio compared to skeletal muscle myosins (*Tables 3* and *4*).

Partial uncoupling of the communication pathway by the conservative R788K mutation has a moderate impact on nucleotide binding and release kinetics (*Table 3*) in the myosin ATPase cycle compared to NM2C. Consequently, the overall kinetic features are similar to NM2C (*Table 3*). Complete uncoupling of the converter from the motor domain in R788E slows down all kinetic steps of the myosin ATPase cycle and decreases its actin-activation, due to altered switch-2 dynamics. The observed kinetic phenotype of the R788E myosin ATPase cycle is similar to the *Dictyostelium* nonmuscle myosin-2 non-hydrolyzer mutants in which the salt bridge between switch-1 and switch-2 is completely destroyed by mutagenesis and unable to form (*Furch et al., 1999*; *Friedman et al., 1998*). Non-hydrolyzer mutants are characterized by long-lived ATP states and reduced nucleotide release and binding kinetics, including a drastic decrease in $k_{+AD}$ (*Furch et al., 1999*; *Friedman et al., 1998*). Like R788E, *Dictyostelium* myosin-2 non-hydrolyzer mutants do not exhibit a nucleotide-induced change in the intrinsic fluorescence signal (*Furch et al., 1999*). It is of note that the lack of an intrinsic fluorescence signal in R788E is caused by the blockage of the communication pathway on the relay helix before or at Y518 (*Figure 6A,B*), whereas the impairment of the nucleotide switches to form a salt bridge and the resulting lack of the switch-2-induced conformational change of the relay helix is the expected cause in the non-hydrolyzer mutants.

F-Actin affinities are largely unaffected by the R788E mutation, which is in line with the observation that switch-1 dynamics are only marginally affected in comparative MD simulations. The presence of F-actin however establishes ATP/ADP selectivity in the actomyosin ATPase cycle: ActoNM2C preferentially binds ADP over ATP, whereas actoR788E preferentially binds ATP over ADP (*Table 3*). The ATP selectivity is caused by an 85-fold decreased second-order ADP-binding rate constant

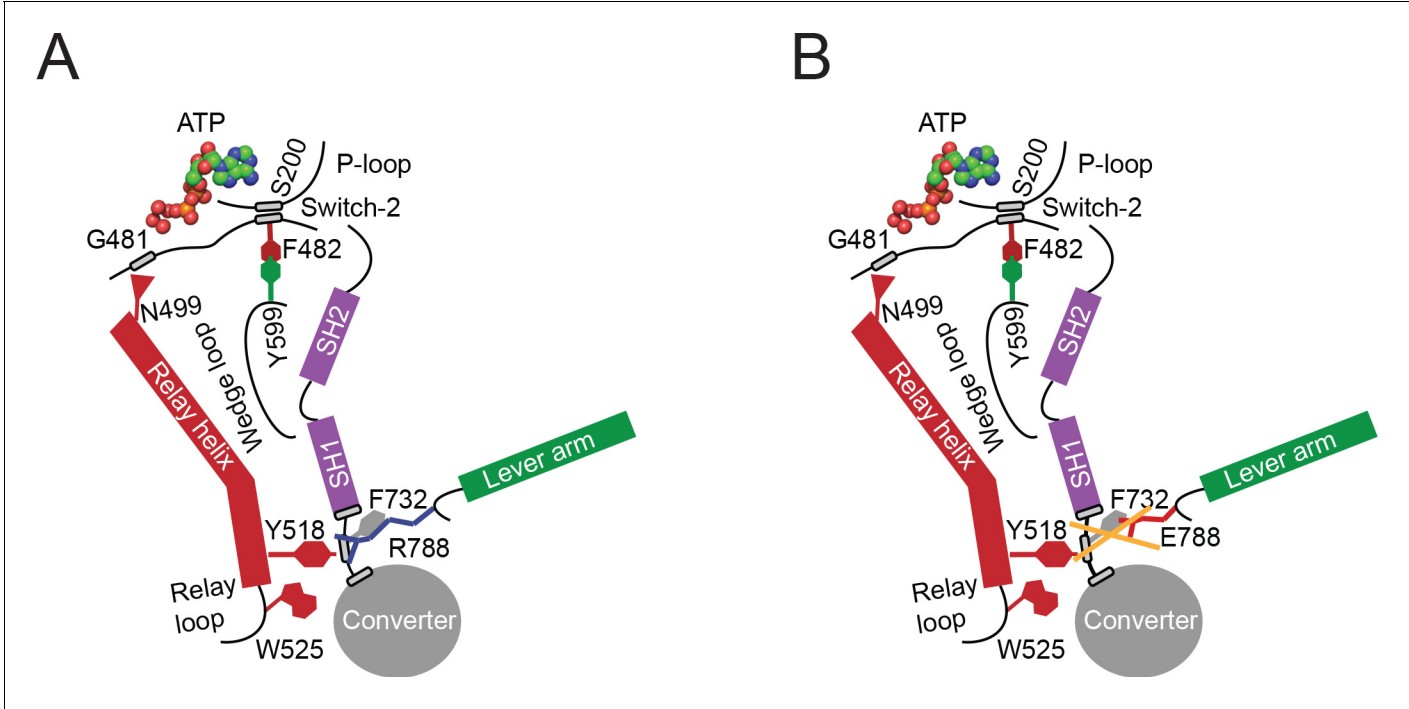

**Figure 6.** Proposed allosteric communication pathway from the converter to the NM2C active site. (**A**) Residue R788 connects the converter to the SH1 interface through main chain and side chain interactions in NM2C. This interface further interacts with Y518 of the relay helix and W525. The interactions are propagated to the active site and *vice versa* through the relay helix and through an interaction of relay helix residue N499 with the main chain of G481 from switch-2. The latter directly interacts with the nucleotide. Y599 of the wedge loop, that itself contacts the relay helix, establishes an interaction with switch-2 F482. This residue is in contact with S200 of the P-loop by main chain interactions and directly interacts with the nucleotide in the active site. Schematic not drawn to scale. (**B**) The interface between R788E and both, the SH1 and the relay helix, is perturbed and the allosteric communication from the active site to the converter compromised. The experimental observation that R788E does not change its intrinsic fluorescence upon nucleotide binding which is caused by a lacking conformational change of W525 indicates that the communication pathway is interrupted in the relay helix. The position of W525 in the relay loop at the distal end of the relay helix indicates that the pathway is interrupted before or at Y518, which is supported by the observation that the relay loop does not change its position during the time course of the MD simulation. As a consequence, Y518 cannot establish an interface with E788 and F732 of the converter and SH1 helix. Further, E788 cannot establish interactions with N776 and V791 and completely disrupts the structural integrity of the interface of converter, SH1-SH2 helix, relay helix and the lever arm and uncouples nucleotide-induced changes in the active site from the converter rotation. Only key residues involved are shown in the proposed mechanism. Small grey boxes indicate main chain interactions. Schematic not drawn to scale.

DOI: https://doi.org/10.7554/eLife.32742.016

$k_{+AD}$, a key signature of R788E actomyosin ATPase cycle. ATP/ADP selectivity is established by actin-induced conformational changes in the myosin motor domain and underlines that the coupling mechanism from the active site to the converter and *vice versa* is different in the presence and absence of F-actin. This observation is in line with recent reports on distinct pathways for the myosin and actin-activated ATPase cycle (*Llinas et al., 2015*).

## Implications for NM2C function in vivo

Nonmuscle myosin-2C assembles into small bipolar filaments that dynamically tether actin filaments in cells (*Ebrahim et al., 2013*; *Billington et al., 2013*). The actomyosin interaction and hence the tension exerted by a sarcomeric array of nonmuscle myosin-2C is of importance for the function and organization of the apical junctional complex in the organ of Corti and actin-rich structures including stress fibers and actin arcs (*Ebrahim et al., 2013*).

Based on our kinetic data, the calculated duty ratio of NM2C suggests that ~14 motor domains would be needed to be geometrically capable of interacting with F-actin at any given time. This number is identical to the number of motor domains per nonmuscle myosin-2C half filament, suggesting that the filament may be at the threshold of being processive in the absence of external loads (*Billington et al., 2013*). It is likely that this threshold is crossed in the presence of other actin

binding proteins, intermolecular loads, and gating between the F-actin bound motor domains (*Hundt et al., 2016*).

The structural prerequisites underlying gating and load-sensitivity in nonmuscle myosins-2 have not been investigated but include a distortion at the converter/lever or the converter/motor domain interface in the nanometer range that is allosterically communicated through the lever arm via the converter to the active site (*Kovács et al., 2007*). Resisting load applied to the lever slows down the actin-activated ADP release from the lead motor of nonmuscle myosin-2A around fivefold thereby increasing the duty ratio, but does not alter the rate of ATP binding (*Hundt et al., 2016*; *Kovács et al., 2007*). The kinetic signatures of the strained nonmuscle myosin-2A lead motor are very similar to the observed kinetic features of R788E. The reduction of the steady-state ATPase of R788E and the concomitant increase in duty ratio and a rate-limiting ADP release rate goes in line with this finding. We propose that internal strain in the nonmuscle myosin-2 dimer distorts the lead motor at the converter/lever interface and leads to its axial translation as seen in electron microscopic studies (*Burgess et al., 2002*). This translation abolishes the interaction of the converter R788 with residues of the SH1-SH2 helix and the relay helix, thereby uncoupling the myosin subdomains and disrupting the communication pathway from the converter to the active site, which is expected to result in a similar kinetic effect as in R788E. Consequently, a motor in a nonmuscle myosin-2 filament would decrease its enzymatic activity and stay strongly bound to F-actin. This feature is required to generate processivity and cytoskeletal tension and of physiological significance in the maintenance of cell shape and tensional homeostasis in the actin cytoskeleton.

## Materials and methods

### Key resources table

| Reagent type (species) or resource | Designation | Source or reference | Identifiers | Additional information |
|---|---|---|---|---|
| Cell line (*Spodoptera frugiperda*) | *Sf9* | Thermo Fisher Scientific | Thermo Fisher Scientific 11496015 | Maintained in Sf-900 III SFM |
| Recombinant DNA reagent | pFastBac1-NMHC-2C0-2R-His$_8$ | PMID: 21478157 | N/A | Progenitors: sequence optimized 1–799 human nonmuscle myosin-2C (GenBank accession number NP_079005) directly fused to spectrin repeats 1 and 2 from Dictyostelium discoideum alpha-actinin; vector pFastBac1 |
| Recombinant DNA reagent | pFastBac1-NMHC-2C0-2R-Flag | this paper | NM2C (kinetic studies) | Progenitor: pFastBac1-NMHC-2C0-2R-His8 |
| Recombinant DNA reagent | pFastBac1-NMHC (45-799)—2C0-2R-His$_8$ | this paper | NM2C (crystal structure) | Progenitor: pFastBac1-NMHC-2C0-2R-His8 |
| Recombinant DNA reagent | pFastBac1-NMHC (R788K)—2C0-2R-Flag | this paper | R788K (kinetic studies) | Progenitor: pFastBac1-NMHC-2C0-2R-Flag |
| Recombinant DNA reagent | pFastBac1-NMHC (R788E)—2C0-2R-Flag | this paper | R788E (kinetic studies) | Progenitor: pFastBac1-NMHC-2C0-2R-Flag |

### Protein production

For structural studies, a His$_8$-tagged human NM2C construct comprising amino acids 45–799 directly fused to spectrin repeats 1 and 2 from α-actinin was generated based on the vector pFastBac1-NMHC-2C0-2R-His$_8$ (*Heissler and Manstein, 2011*). Mutagenesis was accomplished by sequence-specific deletion using a whole-plasmid amplification approach. For kinetic studies, an equivalent motor domain construct comprising amino acids 1–799 of NM2C was cloned into a modified pFast-Bac1 vector containing a cDNA sequence encoding a C-terminal Flag-tag (*Figure 1—figure supplement 1B*). This construct was used as a template to introduce the R788E and R788K mutation by In-Fusion cloning (Clontech, Mountain View, CA 94943, USA). All proteins were recombinantly overproduced in the *Sf9*/baculovirus system, purified to electrophoretic homogeneity, and concentrated to ~10 mg/ml using Vivaspin ultrafiltration units (Sartorius, Göttingen, Germany) as previously described (*Heissler and Manstein, 2011*; *Heissler et al., 2015*). Throughout this manuscript,

numbering refers to the amino acid sequence of full-length nonmuscle myosin-2C (GenBank accession number NP_079005).

## Crystallization of NM2C

NM2C at a concentration of ~10 mg/ml was complexed with the ATP analogue ADP·VO$_4$ and crystallized using the hanging drop vapor diffusion method by mixing 2 µl of protein solution with 2 µl of reservoir solution containing 50 mM Tris pH 8.2, 10% (w/v) PEG-5K MME, 1% (v/v) MPD, and 0.2 M NaCl at 8°C. Rectangular plate shaped crystals grew up to 400 × 300×400 µm$^3$ within 4 weeks. Crystals were soaked in the corresponding mother liquor supplemented with 100 mM NaCl and 20% (w/v) ethylene glycol. Protein crystals were transferred in serial steps of increasing concentrations of the cryo-solution. Crystals were transferred into liquid nitrogen using MiTeGen loops and stored at 100 K until data collection.

## Data collection, processing, and refinement

X-ray diffraction data of NM2C crystals were collected at the beam line BL14.1 at Bessy II (Helmholtz-Zentrum, Berlin, Germany) to 2.25 Å resolution (*Table 1*). Data processing was performed using XDS and SADABS (*Blessing, 1995*). The structure of NM2C in the pre-powerstroke state was solved by molecular replacement using Phaser (*McCoy, 2007*). The crystal structure of the chicken smooth muscle myosin-2 motor domain (PDB entry 1BR4) was used as a search model and the two α-actinin repeats were manually traced and rebuilt. The electron density map was sharpened using Coot (*Emsley and Cowtan, 2004*) to ensure the directionality and identity of the α-helices for the two α-actinin repeats. Maximum likelihood crystallographic refinement was performed using iterative refinement cycles in autoBUSTER (*Bricogne et al., 2011*). Iterative cycles of model building were performed using Coot, and model bias was minimized by building into composite omit maps. The model was initially validated in Coot and final validation was performed using MolProbity (*Davis et al., 2004*).

## Kinetic experiments

The actin-activated ATPase assays under steady-state conditions were performed as described earlier in buffer containing 10 mM MOPS pH 7.0, 50 mM NaCl, 2 mM MgCl$_2$, 2 mM ATP, 0.15 mM EGTA, 40 U/ml l-lactic dehydrogenase, 200 U/ml pyruvate kinase, 200 µM NADH, and 1 mM phosphoenolpyruvate at 25°C with a Cary 60 Bio spectrophotometer (Agilent Technologies, Wilmington, DE) (*Heissler et al., 2015*). Transient state kinetic assays were carried out as described previously with a TgK Hi-tech Scientific SF-61 DX stopped-flow system (TgK Hi-tech Scientific Ltd., Bradford-on-Avon, UK) in SF-buffer (25 mM MOPS pH 7.0, 100 mM KCl, 5 mM MgCl$_2$ and 0.1 mM EGTA) unless stated otherwise (*Heissler and Manstein, 2011*). Initial data fitting was performed with Kinetic Studio Version 2.28 (TgK Hi-tech Scientific Ltd., Bradford-on-Avon, UK). Plots were generated with OriginPro 8.5 (OriginLab Corp., Northampton, MA). Data interpretation is according to the kinetic scheme of the myosin and actomyosin ATPase cycle as presented in *Figure 1—figure supplement 1A*.

## Fluorescence measurements

Tryptophan fluorescence emission spectra of 4 µM NM2C or R788E in the presence and absence of 0.5 mM ADP and 0.5 mM ATP, respectively, were measured after excitation at 297 nm at a temperature of 20°C in a QuantaMaster fluorescence spectrophotometer (Photon Technology International, Birmingham, NJ) as described previously (*Málnási-Csizmadia et al., 2007*). Prior to the assay, proteins were transferred to SF-buffer with zebra spin desalting columns (Thermo Fisher Scientific GmbH, Dreieich, Germany). The fluorescence was normalized to the maximum fluorescence of NM2C or R788E in the absence of nucleotide.

## Molecular dynamics simulations

Molecular dynamics based in silico site directed mutagenesis and simulations were performed using NAMD 2.9 and the CHARMM27 force field (*MacKerell et al., 1998*; *Phillips et al., 2005*). The X-ray crystal structure of the NM2C motor domain in the pre-powerstroke state, encompassing residues 49–807, served as the starting structure for NM2C, R788K, and R788E simulations. Mutations were

introduced in silico and the proteins were prepared and optimized prior to MD simulations using the Protein Preparation Wizard of the Schrödinger software suite (Schrödinger Suite 2012 Protein Preparation Wizard; Epik version 2.3; Impact version 5.8; Prime version 3.1; Maestro version 9.3. Schrödinger LLC, New York, NY). The proteins were fully hydrated with explicit solvent using the TIP3P water model and charge neutralization was accomplished by adding $Na^+$ counter ions (*Jorgensen et al., 1983*). Simulations for NM2C, R788K, and R788E were performed in the presence of ATP in the active site. Short-range cutoffs of 12 Å were used for the treatment of non-bonded interactions; while long-range electrostatics was treated with the particle-mesh Ewald method (*Darden and Pedersen, 1993*). All simulations were carried out in an NpT ensemble at 310 K and 1 atm using Langevin dynamics and the Langevin piston method. A 1 fs time step was applied. Prior to production runs the solvated systems were subjected to an initial energy minimization and subsequent equilibration of the entire system for 5 to 10 ns. All MD simulations were carried out at the Computer Cluster of the Norddeutscher Verbund für Hoch- und Höchstleistungsrechnen.

## Acknowledgements

We thank the staff at the beamline BL14-1 at BESSY for technical support. We thank the Norddeutscher Verbund für Hoch- und Höchstleistungsrechnen (HLRN) for providing computational resources and the Biophysics Core of the National Heart, Lung, and Blood Institute (NHLBI) for advice, support and the use of the facility. Data deposition: The atomic coordinates and structure factors have been deposited in the Protein Data Bank, www.pdb.org (PDB entry 5I4E).

## Additional information

### Funding

| Funder | Grant reference number | Author |
| --- | --- | --- |
| Deutsche Forschungsgemeinschaft | MA 1081_21-1 | Dietmar J Manstein |
| National Institutes of Health | Intramural Funding | James R Sellers |
| Deutsche Forschungsgemeinschaft | PR 1478_2-1 | Matthias Preller |

The funders had no role in study design, data collection and interpretation, or the decision to submit the work for publication.

### Author contributions

Krishna Chinthalapudi, Data curation, Formal analysis, Validation, Investigation, Visualization, Methodology, Writing—original draft, Writing—review and editing, Designed experimental approaches, Performed the crystallization, Collected the diffraction data and solved the structure; Sarah M Heissler, Formal analysis, Validation, Investigation, Visualization, Methodology, Writing—original draft, Writing—review and editing, Designed experimental approaches, Cloned, expressed and purified the proteins, Performed the biochemical and kinetic assays; Matthias Preller, Formal analysis, Validation, Investigation, Visualization, Methodology, Writing—original draft, Writing—review and editing, Designed experimental approaches, Performed and analyzed the molecular dynamics simulations; James R Sellers, Supervision, Funding acquisition, Validation, Writing—original draft, Writing—review and editing, Designed experimental approaches; Dietmar J Manstein, Conceptualization, Resources, Data curation, Software, Formal analysis, Supervision, Funding acquisition, Validation, Methodology, Writing—original draft, Project administration, Writing—review and editing, Conceived the study and designed experimental approaches

### Author ORCIDs

Matthias Preller http://orcid.org/0000-0002-7784-4012
James R Sellers https://orcid.org/0000-0001-6296-564X
Dietmar J Manstein http://orcid.org/0000-0003-0763-0147

Decision letter and Author response
Decision letter https://doi.org/10.7554/eLife.32742.020
Author response https://doi.org/10.7554/eLife.32742.021

# Additional files

## Supplementary files

• Supplementary file 1. Dihedral angles of switch-1 and lever arm residues in crystal structures of NM2C and smooth muscle myosin-2.
DOI: https://doi.org/10.7554/eLife.32742.017

• Transparent reporting form
DOI: https://doi.org/10.7554/eLife.32742.018

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
