## [Decision Letter]

[Editors’ note: a previous version of this study was rejected after peer review, but the authors submitted for reconsideration. The first decision letter after peer review is shown below.]

Thank you for submitting your work entitled "Mechanistic Insights into the Active Site and Allosteric Communication Pathways in Human Nonmuscle Myosin-2C" for consideration by *eLife*. Your article has been reviewed by three peer reviewers, one of whom, Pekka Lappalainen (Reviewer #1), is a member of our Board of Reviewing Editors, and the evaluation has been overseen by a Senior Editor.

Our decision has been reached after consultation between the reviewers. Based on these discussions and the individual reviews below, we regret to inform you that your work will not be considered further for publication in *eLife*.

All three reviewers found the topic of this study very important and appreciated that the manuscript presents a novel, high-quality pre-powerstroke structure of non-muscle myosin-IIC. However, they also stated that, despite predictions made from the novel structural features of myosin-IIC, this study does not provide direct experimental insights into how these structural adaptations result in non-muscle myosin-2 specific kinetic signatures. Moreover, reviewer #3 raised some technical concerns about the kinetics experiments.

Thus, an extensive amount of additional mutagenesis and biochemistry work would be required to reveal how structural adaptations lead to characteristic kinetic features of non-muscle myosin-IIC. Because our policy is that a revision is invited only when it can be carried out in 2-3 months, we cannot unfortunately offer to publish this work in *eLife.*

Reviewer #1:

The precise molecular mechanisms underlying actin-myosin force generation, as well as the principles by which different myosins are kinetically fine-tuned for specific cellular functions, are incompletely understood. Here, Chinthalapudi et al. report a high-resolution pre-powerstroke state crystal structure of the human non-muscle myosin-IIC (NM2C) motor domain, and applied mutagenesis and molecular dynamics simulation approaches to elucidate the molecular basis of the characteristic kinetic features of this protein. Through careful structural analysis, they revealed interesting differences in the active sites and interdomain connectivity between NM2C and fast sarcomeric myosin-II. Mutagenesis experiments demonstrated that a conserved arginine (R788) in NM2C is critical for allosteric communication between the active site and the distal end of the motor domain.

This study provides interesting new information concerning the interplay between the active site and the converter/Nter/lever interface of non-muscle myosin-II. However, in its present form the manuscript does not provide experimental insights into how structural adaptations in the active site and interdomain connectivity result in specific properties of NM2C. Thus, the study would be significantly stronger, if the authors would perform more extensive mutagenesis analysis to elucidate the roles of non-muscle myosin II -specific structural features in the characteristic kinetic features of these proteins as compared to fast sarcomeric myosin II.

Major comments:

The authors disrupted the allosteric communication pathway between the distal end and the active site through replacing the arginine-788 by glutamate. Analysis of the mutant protein nicely demonstrates the importance of this residue in the interplay between the active site and the distal end of NM2C. However, in my opinion analysis of the R778E mutant does not provide insights into non-muscle myosin-2 specific kinetic signatures, as stated in the Abstract and in the Discussion. Thus, the impact of this study would be significantly increased if the authors could perform more extensive structure-guided mutagenesis around the active site and at the 'connecting hub' with the aim of converting a sarcomeric myosin II to a more NM2C-like, or vice versa. Based on the extensive structural analysis presented, the authors already suggest that the differences e.g. in the length of JK-loop, and connections between the JK-loop, switch-1 and the nucleotide may explain the distinct kinetic features of muscle and non-muscle myosin-II, and these could perhaps be tested by mutagenesis.

Reviewer #2:

This paper presents one more high resolution structure of a myosin II with ADP and vanadate bound in the active site giving a pre-power stroke conformation of the motor domain. This is a non-muscle myosin the 2c isoform which has roles in cytokinesis, neuronal dynamics adhesion and tension maintenance. So the question is what is of interest about one more myosin II structure?

In fact this paper uses the new 2.5 Å structure to launch into a very detailed, carefully argued case for what makes this particular isoform different to the other previously examined isoforms. In doing so the paper raises a series of very important topics – namely why are there so many distinct myosin II isoforms and what gives the motor domain its specific mechanical properties that match it its biological role? How is the structure modified to generate this particular set of properties? The structure highlights two areas a loop between Helix J & K which is shortened in M2C and a key residue in the converter region, R788, which has key interactions with the lever arm, the SH1 helix and the relay helix.

The shortened loop J/K loop disrupts the nucleotide pocket and decouples actin and ADP binding. R788 is a part of the link between the nucleotide pocket and the lever arm movement. Disruption of this in the R788E mutation results in a dramatic change in motor properties specifically in the *K_AD_/K_D_* ratio and the duty cycle.

My only criticism of the paper is a limitation of the R788E mutation used. This charge is a change and is expected to cause major disruption of the local structure and the connectivity of the converter domain with the other local domains. The prediction is that the more subtle change, R778K, as seen in the striated muscle myosins, would have also had a significant effect on motor properties without such a dramatic local charge change. Had the authors considered this mutation as a test of their model?

This is an important paper that addresses fundamental issues of how myosins are designed for motor specific functions and as such will be of interest to the myosin community and to those with interests in protein structure-function relationships.

Reviewer #3:

The paper by Chinthalapudi et al. reports a new pre-powerstroke structure of non-muscle myosin-IIC. Novel features of this structure includes the conformation of the JK helix, a region that provides a connection between switch-1 and the nucleotide, the switch-1 conformation, and the lever position (Table 2). Predictions are made regarding how minor structural features may result in kinetic differences in the ATPase (Table 2); however, data testing these predictions are minimal. Rather, the major focus of the paper is a "hub region" that includes R788, which appears to be important for the allosteric pathway linking the motor lever am and active site. The hub region is not novel to this myosin, and R788 is conserved among myosin-IIs and members of the myosin superfamily (although residues numbers are different). Given its conservation and clear interactions among key subdomain, it is not surprising that R788 is an important allosteric connection.

I am concerned that the authors are tracing allosteric pathways and performing kinetic experiments with a myosin construct missing key subdomains. It has been shown previously that removal of the lever arm and/or essential-light-chain (ELC) impacts several of the steps on the ATPase pathway. Indeed, the ELC interacts with the motor domain in the pre-power-stroke state (e.g., 1BR1). Additionally, the Sutoh and Manstein group showed in 2006 that mutations with the N-terminus of the myosin have substantial effects on kinetic parameters. Therefore, one has to question if the usefulness of the molecular dynamics simulations and tracing of allosteric pathways with this modified construct.

[Editors’ note: what now follows is the decision letter after the authors submitted for further consideration.]

Thank you for submitting your article "Mechanistic Insights into the Active Site and Allosteric Communication Pathways in Human Nonmuscle Myosin-2C" for consideration by *eLife*. Your article has been reviewed by three peer reviewers, one of whom, Pekka Lappalainen (Reviewer #1), is a member of our Board of Reviewing Editors, and the evaluation has been overseen by Anna Akhmanova as the Senior Editor. The following individual involved in review of your submission has agreed to reveal their identity: Michael Geeves (Reviewer #2).

The reviewers have discussed the reviews with one another and the Reviewing Editor has drafted this decision to help you prepare a revised submission.

The structural and mechanistic details underlying diversity in the enzymatic properties and physiological functions between the members of myosin-2 protein family are incompletely understood. This is a revised version of a manuscript that reports a crystal structure of the pre-powerstroke state of human non-muscle myosin 2C (NM2C). Through a thorough analysis of the structure, combined with mutagenesis, biochemistry and MD simulations, the authors reveal a communication pathway that connects the active site and the distal end of the motor domain, and provide evidence that this pathway contributes to NM2C-specific kinetic properties. Most importantly, in the revised manuscript the authors present analysis of a less drastic mutation in the converter domain of NM2C R788K vs R788E. This does significantly strengthen the arguments for the role of this residue in the allosteric communication between the nucleotide pocket and the converter domain.

The three reviewers found the manuscript significantly improved, and stated that it provides valuable new information on the interplay between the active site and the converter/Nter/lever interface of non-muscle myosin-II. However, the manuscript would be much stronger if the authors could edit the Results and Discussion sections to better convey the most important findings of the study, and to make it accessible to a wider audience. The reviewers agree that many of the atomistic details presented in the paper are important to myosin specialists, and they should be included in the Supplement.

---

## [Author Response]

[Editors’ note: the author responses to the first round of peer review follow.]

Reviewer #1:[…] This study provides interesting new information concerning the interplay between the active site and the converter/Nter/lever interface of non-muscle myosin-II. However, in its present form the manuscript does not provide experimental insights into how structural adaptations in the active site and interdomain connectivity result in specific properties of NM2C. Thus, the study would be significantly stronger, if the authors would perform more extensive mutagenesis analysis to elucidate the roles of non-muscle myosin II -specific structural features in the characteristic kinetic features of these proteins as compared to fast sarcomeric myosin II.

The emphasis of this work is to elucidate how nonmuscle myosins-2 adopted their kinetic signatures that are very distinct from their sarcomeric myosin counterparts. It does not aim to address with extensive experimental studies how mutations in the active site contribute to NM2C specific features. Importantly, active site mutations have been extensively studied in nonmuscle myosins-2 (Nat Cell Biol. 2001 Mar;3(3):311-5; Adv Biophys. 1998;35:1-24; J Biol Chem. 1998 Oct 16;273(42):27404-11; Biochemistry. 1997 Nov 18;36(46):14037-43. J Mol Biol. 1999 Jul 16;290(3):797-809; Biochemistry. 1998 Jul 7;37(27):9679-87) and often result in the complete loss of the enzymatic activity so that no firm conclusions on allosteric pathways downstream the active site can be drawn. Also, the replacement of surface loops with the respective region from other myosins only yielded inconclusive results (J Biol Chem. 1998 Mar 13;273(11):6262-70; Biochem Biophys Res Commun. 2008 Apr 25;369(1):124-34; J Biol Chem. 1998 Oct 23;273(43):27939-44; J Biol Chem. 1995 Dec 22;270(51):30260-3) and failed to convert one myosin into another.

While we did not address active site mutations, we did explore the effect of single point mutations of residue R788 on the interdomain connectivity of NM2C. This involved extensive experimental and in silico studies. To address the reviewer’s concern in the revised version of the manuscript, we include a detailed analysis of the impact of mutation R788K on the interdomain connectivity at the converter/Nter/lever junction on NM2C activity by a combined molecular dynamics and kinetic characterization.

We strongly agree with the reviewer that it would be a breakthrough in myosin research, if we could convert a slow myosin like nonmuscle myosin-2C into a fast myosin like sarcomeric myosin-2 by the introduction of a single point mutation. However, several lines of previous research argue against the feasibility of this approach, as outlined in greater detail below.

Major comments:The authors disrupted the allosteric communication pathway between the distal end and the active site through replacing the arginine-788 by glutamate. Analysis of the mutant protein nicely demonstrates the importance of this residue in the interplay between the active site and the distal end of NM2C. However, in my opinion analysis of the R778E mutant does not provide insights into non-muscle myosin-2 specific kinetic signatures, as stated in the Abstract and in the Discussion. Thus, the impact of this study would be significantly increased if the authors could perform more extensive structure-guided mutagenesis around the active site and at the 'connecting hub' with the aim of converting a sarcomeric myosin II to a more NM2C-like, or vice versa. Based on the extensive structural analysis presented, the authors already suggest that the differences e.g. in the length of JK-loop, and connections between the JK-loop, switch-1 and the nucleotide may explain the distinct kinetic features of muscle and non-muscle myosin-II, and these could perhaps be tested by mutagenesis.

We are just at the very beginning of a process that aims to elucidate the allosteric pathways that communicate molecular events from the nucleotide binding site, the actin-binding region and the distal end of the converter domain to generate force and movement on F-actin. The myosin motor domain comprises around 800 amino acids. Out of these, around 400 are identical between human NM2C (NP_079005.3)) and skeletal muscle myosin-2 (AAI14546.1). It is outside the scope of the present study to mutate a rationally selected subset, or combinations of these residues with the aim to convert a slow into a fast type 2 myosin. Another complication with the suggested experiments is that – for a proof of principle – a fast skeletal muscle myosin-2 should be converted by a point mutation into a slow myosin-2. Muscle myosins-2 motor constructs cannot be overproduced in the quantities needed for thorough kinetic and structural studies due to folding problems in the baculovirus/Sf9 insect cell system. Adenoviral expression systems (Nat Commun. 2015 Aug 6;6:7974; Cell Mol Life Sci. 2012 Jul;69(13):2261-77; J Biol Chem. 2013 Sep 20;288(38):27469-79) can be used instead but give more than 10-fold lower yields (J Biol Chem. 2013 Sep 20;288(38):27469-79) compared to NM2C produced in of Sf9 insect cells. These limitations underline that the type of experiments the reviewer suggests are not feasible at this point and would require sophisticated high throughput approaches and potentially years of additional work.

To address some of the concerns raised by the reviewer, we included the detailed analysis of mutant R788K in the revised version of the manuscript. The combined molecular dynamics and kinetic characterization of mutants R788K and R788E allowed us to identify residues that are involved in the allosteric communication from the active site to the distal end of the myosin motor domain and helped us expanding our discussion on the allosteric communication in the myosin motor domain.

Reviewer #2:[…] My only criticism of the paper is a limitation of the R788E mutation used. This charge is a change and is expected to cause major disruption of the local structure and the connectivity of the converter domain with the other local domains. The prediction is that the more subtle change, R778K, as seen in the striated muscle myosins, would have also had a significant effect on motor properties without such a dramatic local charge change. Had the authors considered this mutation as a test of their model?This is an important paper that addresses fundamental issues of how myosins are designed for motor specific functions and as such will be of interest to the myosin community and to those with interests in protein structure-function relationships.

We thank the reviewer for highlighting the significance of our work for the understanding of structure-function relationship in nonmuscle myosins-2. Though we have indirectly shown how the lysine residue at the equivalent position to R788 establishes interconnectivity between the Nter, the converter and the lever in sarcomeric myosins-2 (Figure 3, Figure 3—figure supplement 1), we now produced mutant R788K as suggested by the reviewer and included the detailed steady-state and transient kinetic description in the manuscript. We initially generated the charge-reversal mutant R788E to purposefully disrupt side chain:side chain as well as side chain:main chain interactions. We expected mutant R788E to have a moderate effect since it does not disrupt main chain:main chain interactions. In mutant R788K, most of the side chain:side chain as well as side chain:main chain interactions are preserved. Accordingly, mutant R788K displays only mild changes in its kinetic behavior. The results suggest a graded effect of the substituted residue on myosin motor function. We also include the key findings from a 100 ns molecular dynamics simulation in the manuscript, to allow for a direct comparison of the dynamic features of both mutants compared to the wild type protein.

Reviewer #3:The paper by Chinthalapudi et al. reports a new pre-powerstroke structure of non-muscle myosin-IIC. Novel features of this structure includes the conformation of the JK helix, a region that provides a connection between switch-1 and the nucleotide, the switch-1 conformation, and the lever position (Table 2). Predictions are made regarding how minor structural features may result in kinetic differences in the ATPase (Table 2); however, data testing these predictions are minimal. Rather, the major focus of the paper is a "hub region" that includes R788, which appears to be important for the allosteric pathway linking the motor lever am and active site. The hub region is not novel to this myosin, and R788 is conserved among myosin-IIs and members of the myosin superfamily (although residues numbers are different). Given its conservation and clear interactions among key subdomain, it is not surprising that R788 is an important allosteric connection.

We respectfully disagree with the statement that the described “hub region” is not novel to this myosin. To our best knowledge, no study has addressed the contribution of residues in this region with elements of the Nter, lever, and converter in NM2C. Moreover, the residue corresponding to R788 has not been studied in any other nonmuscle myosins-2, including the very well characterized myosin-2 from the model organism *Dictyostelium* that has been subject to several dozens of kinetic and mutagenesis studies. This is not a surprise, since the interaction between elements of the myosin motor domain and the converter is not obvious in crystal structures of smooth and sarcomeric myosins-2 (Figure 3—figure supplement 1).

I am concerned that the authors are tracing allosteric pathways and performing kinetic experiments with a myosin construct missing key subdomains. It has been shown previously that removal of the lever arm and/or essential-light-chain (ELC) impacts several of the steps on the ATPase pathway. Indeed, the ELC interacts with the motor domain in the pre-power-stroke state (e.g., 1BR1). Additionally, the Sutoh and Manstein group showed in 2006 that mutations with the N-terminus of the myosin have substantial effects on kinetic parameters. Therefore, one has to question if the usefulness of the molecular dynamics simulations and tracing of allosteric pathways with this modified construct.

We respectfully disagree with the general statement that the removal of the light chain binding region in nonmuscle myosins-2 impacts the ATPase pathway, since these statements apply only to double headed myosin constructs. In single headed myosin constructs, the ELC-mediated effects are lost. The Manstein and other laboratories have proven this point again and again over the last 20 years and reading a statement to the opposite by referees who provide no proof or reference supporting their opinion is not justified. Our laboratories studied non-muscle myosin-2 and other myosin in their full-length, double- and single-headed forms in great detail. Before publishing a number of high-profile papers on the re-engineering of the myosin lever-arm domain, we demonstrated that (if performed in the correct way!) the replacement of the light chain binding region by an artificial lever arm does not at all affect myosin motor domain kinetics (1), motor function (2) and structure (3). We demonstrated the validity of this approach again when we started our work on NM-2C (4). The latter paper contains the following statement:

“The concept of fusing the myosin motor domain to an artificial lever arm facilitates the characterization of the motor properties and was successfully demonstrated for myosins of different classes from various species in cellular, mechanical, and kinetic studies. Light chain binding domain-mediated regulation is observed only with double-headed constructs. When double-headed constructs are used, the light chain binding domain of NMHC-2C has been observed to exert an inhibitory effect on isoforms lacking the C2 insert. In the case of isoform NMHC-2C1, the inhibition mediated by the light chain binding domain needs to be overcome by MLC20 phosphorylation for both maximum actin-activated Mg-ATPase activity and maximum in vitro motility (21). Therefore, the rate constants and motile activity observed with the singleheaded motor domain constructs with artificial lever arm used here reflected the activity of the fully activated enzyme.”

Accordingly, our manuscript submitted to *eLife* deals with the properties of the fully activated NM-2C motor and its conformational dynamics during ATP turnover. Not more and not less.

1) Kurzawa, S. E., D. J. Manstein, and M. A. Geeves. 1997. Dictyostelium discoideum myosin II: characterization of functional myosin motor fragments. Biochemistry 36:317-323;

2) Ruff, C., M. Furch, B. Brenner, D. J. Manstein, and E. Meyhofer. 2001. Single-molecule tracking of myosins with genetically engineered amplifier domains. Nat Struct Biol 8:226-229.)

3) Kliche, W., S. Fujita-Becker, M. Kollmar, D. J. Manstein, and F. J. Kull. 2001. Structure of a genetically engineered molecular motor. Embo J 20:40-46.

4) Heissler, S. M., and D. J. Manstein. 2011. Comparative kinetic and functional characterization of the motor domains of human nonmuscle myosin-2C isoforms. J Biol Chem 286:21191-21202.

The Sutoh and Manstein paper did explore the effect of the N-terminus/SH3-like domain on the enzymatic activity of a motor domain construct of *Dictyostelium* nonmuscle myosin-2. In case of NM2C, all studied constructs contain the conserved SH3-like domain and the first 44 amino acids prior to the SH3-like domain were truncated for crystallization only (Figure 1—figure supplement 1). Truncation of the amino acids did not change key parameters of the myosin kinetic cycle and the obtained crystal structure only shows very small root mean square deviations (0.57-0.78 Å) when compared to previous structures of smooth, striated, and nonmuscle myosins-2 (Figure 1—figure supplement 1), indicating that the truncation of these residues does not perturb folding and the overall conformation of the myosin motor domain.

The molecular dynamics simulations were performed on the NM2C motor domain only and do not include the artificial lever arm as clearly stated in the Material and methods section of the manuscript. Molecular dynamics simulations on myosin motor domain constructs without the light chain binding region/light chains is standard in the field (Proc Natl Acad Sci U S A. 2010 Mar 16;107(11):5001-5; PLoS Comput Biol. 2007 Feb 16;3(2):e23; Proc Natl Acad Sci U S A. 2011 May 10;108(19):7793-8; Proc Natl Acad Sci U S A. 2005 May 10;102(19):6873-8; Structure. 2007 Jul;15(7):825-37). Moreover, densities for the N-terminal region of the cocrystallized light chains are not resolved in most myosin structures.

[Editors' note: the author responses to the re-review follow.]

[…] The three reviewers found the manuscript significantly improved, and stated that it provides valuable new information on the interplay between the active site and the converter/Nter/lever interface of non-muscle myosin-II. However, the manuscript would be much stronger if the authors could edit the Results and Discussion sections to better convey the most important findings of the study, and to make it accessible to a wider audience. The reviewers agree that many of the atomistic details presented in the paper are important to myosin specialists, and they should be included in the Supplement.

We thank the reviewers for the constructive and helpful review of our work “Mechanistic Insights into the Active Site and Allosteric Communication Pathways in Human Nonmuscle Myosin-2C”. We are pleased by the reviewer’s positive response to the revised version of our manuscript and included a point-by-point reply to the comments. We have included new data for the MD simulations of R788K in Figure 5—figure supplement 1 and edited the Results and Discussion in the manuscript to address the issues raised by the reviewers.

We are slightly surprised by the suggestion to move the atomistic details presented in our work to the supplement – especially since the reviewers previously suggested to include more atomistic details. However, we edited the Results and Discussion as suggested by the reviewers to improve readability and included two Supplementary files in the revised version of the manuscript.